# Nanoengineering of cathode layers for solid oxide fuel cells to achieve superior power densities

Katherine Develos-Bagarinao [1✉], Tomohiro Ishiyama [2], Haruo Kishimoto[1], Hiroyuki Shimada [3] & Katsuhiko Yamaji[2]

Solid oxide fuel cells (SOFCs) are power-generating devices with high efficiencies and considered as promising alternatives to mitigate energy and environmental issues associated with fossil fuel technologies. Nanoengineering of electrodes utilized for SOFCs has emerged as a versatile tool for significantly enhancing the electrochemical performance but needs to overcome issues for integration into practical cells suitable for widespread application. Here, we report an innovative concept for high-performance thin-film cathodes comprising nanoporous $La_{0.6}Sr_{0.4}CoO_{3-\delta}$ cathodes in conjunction with highly ordered, self-assembled nanocomposite $La_{0.6}Sr_{0.4}Co_{0.2}Fe_{0.8}O_{3-\delta}$ (lanthanum strontium cobalt ferrite) and $Ce_{0.9}Gd_{0.1}O_{2-\delta}$ (gadolinia-doped ceria) cathode layers prepared using pulsed laser deposition. Integration of the nanoengineered cathode layers into conventional anode-supported cells enabled the achievement of high current densities at 0.7 V reaching ~2.2 and ~4.7 A/cm$^2$ at 650 °C and 700 °C, respectively. This result demonstrates that tuning material properties through an effective nanoengineering approach could significantly boost the electrochemical performance of cathodes for development of next-generation SOFCs with high power output.

[1] Global Zero Emission Research Center, National Institute of Advanced Industrial Science and Technology (AIST), Tsukuba, Ibaraki, Japan. [2] Research Institute for Energy Conservation, National Institute of Advanced Industrial Science and Technology (AIST), Tsukuba, Ibaraki, Japan. [3] Innovative Functional Materials Research Institute, National Institute of Advanced Industrial Science and Technology (AIST), Nagoya, Aichi, Japan. ✉email: develos-bagarinao@aist.go.jp

Solid oxide fuel cells (SOFCs) have attracted considerable research interest[1–3] due to their high efficiencies in converting chemical energy to electricity as well as their flexibility in utilizing conventional hydrocarbon-based fuels such as methane and ammonia, and in recent years have seen a surge in the research and deployment worldwide of commercial systems ranging from stationary to transport applications. Durability and performance stability over long-term operation have been the focus of R&D efforts in both academic and industrial sectors[4–11], however, a persistent key issue required for widespread commercialization of SOFC technology is the lowering of the system cost. In addition to improving production routes and efficiency of cell stacks, improving performance by achieving higher power densities has been identified as a strategy to reduce stack size and system cost. Toward this purpose, advanced thin-film techniques such as pulsed laser deposition (PLD) have been utilized to explore alternative and nanoengineered cathode materials exhibiting high performance in terms of low area-specific resistance (ASR) and high oxygen exchange properties superior to those of conventional cathodes prepared by screen-printing techniques[12–17]. These studies indicate that the functionalities of existing cathode materials can be engineered at the nanoscale to specifically tailor properties required for applications. Successful implementation using the non-vacuum-based sol-gel technique has also been reported to obtain record-low values of the ASR for nanoscaled LSC ($La_{0.6}Sr_{0.4}CoO_{3-\delta}$) cathodes[18]. Nevertheless, most studies have so far only focused on the initial performance but not the long-term stability, which is considered to be one of the most crucial issues hindering the application of thin-film cathodes in practical cells. Though maintaining the stability of thin-film cathodes undoubtedly presents unique challenges as compared to conventional porous cathodes, addressing such challenges would be beneficial in developing guidelines for future fabrication processes and cell evaluation techniques.

One promising approach is to combine a conventional cathode material such as LSC or LSCF (lanthanum strontium cobalt ferrite, $La_{0.6}Sr_{0.4}Co_{0.2}Fe_{0.8}O_{3-\delta}$) perovskite oxide with a good ionic conductor such as rare-earth-doped ceria to form nanocomposite structures[19]. This has the benefit of enhancing the cathode/electrolyte interfacial area density and facilitating the oxide ion transfer, leading to a significant decrease of the polarization resistance values. It must be noted that to date only LSC-GDC (gadolinia-doped ceria, $Ce_{0.9}Gd_{0.1}O_{2-\delta}$)[19] and SSC-SDC (Sr-doped $SmCoO_3$ and Sm-doped $CeO_2$)[20] nanocomposites have so far been reported to be successfully grown via PLD, but no similar study has yet reported the successful fabrication of nanocomposite thin films comprising LSCF and rare-earth-doped ceria. Another critical issue is how to tailor cathode nanostructures and retain active sites for oxygen reduction even at high operating temperatures. Thin-film cathodes typically suffer from an inevitable loss of nanostructures when subjected to high temperatures due to thermally induced grain coarsening[21] and surface segregation[22–25] occurring at such conditions, leading to significant degradation of the oxygen surface exchange properties. In addition, due to the intrinsically poor lateral conductivity of thin-film cathodes arising from the characteristic columnar microstructure, the cell performance of these materials has not yet been fully optimized in terms of appropriately selecting current collectors which would enable good electrical contact with such nanometer-sized grains[26,27].

The goal of this study is to develop advanced cathode materials with superior performance compared to conventional cathodes for the development of next-generation SOFCs. Here, we implement an innovative cell architecture utilizing at its core nanoengineered cathode layers comprising self-assembled LSCF and GDC nanocomposite thin films in conjunction with nanoporous LSC thin films. The nanoscale distribution of LSCF and GDC phases in the dense nanocomposite performs as a highly efficient transition layer by providing a high interfacial density and good adhesion with the underlying GDC interlayer, whereas the LSC thin film with its nanoporous, open structure ensures high surface area for the oxygen reduction reaction. Next, the integration of the nanoengineered cathodes in commercially viable Ni-YSZ (yttria-stabilized zirconia) anode-supported cells is demonstrated, where the roles of the respective layers in the cathode structure on the electrochemical performance are systematically examined. In the optimized cell configuration, superior electrochemical performance with high current densities at 0.7 V reaching ~2.2 and ~4.7 A/cm$^2$ at 650 and 700 °C, respectively, are achieved.

## Results

**LSCF-GDC nanocomposite layer**. We first focus our attention on the development of the nanocomposite layer, which forms the initial layer in direct contact with the electrolyte. To function as an effective transition layer at the electrode–electrolyte interface, nanocomposites should ideally have a fine, nanoscale-level distribution of the two phases to significantly increase the interfacial density as well as exhibit good adhesion with the electrolyte. Through an appropriate selection of PLD parameters, the resulting nanocomposite film can be finely tuned to have a nanoscale-level distribution of the LSCF and GDC phases. The LSCF-GDC nanocomposite films are revealed to be highly ordered and self-assembled in the nanoscale, as evidenced by the typical low-magnification STEM-HAADF (scanning transmission electron microscopy/high-angle annular dark-field) image (Fig. 1a) of a LSCF-GDC nanocomposite film (~300 nm) prepared on a GDC substrate. Here, the nanocomposite film is comprised of granular domains in the order of tens of nanometers; additionally, each domain appears to contain alternately arranged LSCF and GDC nanostripes with widths in the range of ~2–5 nm within each individual granular domain. The nanostripes exhibit long-range ordering throughout an individual granular domain spanning its entire thickness and are either oriented along the normal to the GDC substrate surface or tilted at an angle. The preferentially tilted growth direction can be ascribed to the 60° incident angle of the laser beam in the design of the PLD system (as opposed to the conventional 45° configuration), such that the ablated species from the target would arrive on the substrate from a slightly oblique direction. The nanostripes appear more clearly at the top region of the nanocomposite film compared to those near the interface with GDC, however, this is more likely attributed to the FIB milling process than any particular change in growth mode (due to the ion beam broadening effects, the ion-milled specimen is usually thinner at the top than at the bottom and consequently exhibits better electron transparency). Assuming that the individual phases are distributed uniformly and equally in domains of ~5 nm each within a ~300-nm thickness, this unique self-assembled microstructure effectively increases the cathode/electrolyte interfacial area density by as high as 100 times or more, compared to a baseline sample of the same unit area where the only interface exists between two adjacent layers comprising a pure LSCF layer and GDC electrolyte.

Figure 1b shows the STEM-EDX (energy-dispersive X-ray spectroscopy) elemental mapping obtained from the area denoted by a dashed rectangle on Fig. 1a. Due to the overlapping of the La peaks with Ce in the EDX spectra, the La mapping exhibits the same pattern as that of Ce. Nevertheless, all other elements composing LSCF, namely, Sr, Co, and Fe, congruently appear in

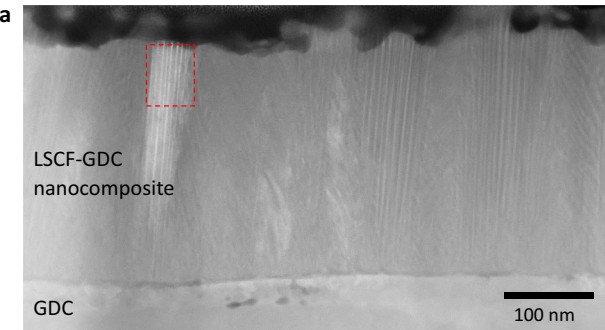

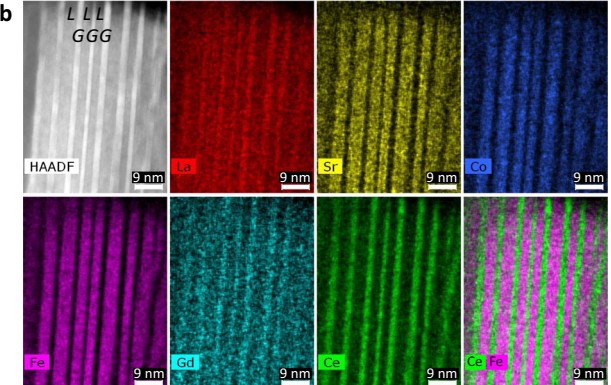

**Fig. 1 Microstructure and phase distribution of the LSCF-GDC nanocomposite film. a** Low-magnification STEM-HAADF image showing the existence of self-assembled nanostriped patterns with long-range ordering across the thickness of the LSCF-GDC nanocomposite film. **b** STEM-HAADF and STEM-EDX elemental distribution of the various elements comprising LSCF (La, Sr, Co, Fe) and GDC (Gd, Ce) phases within the area denoted by a dashed rectangle in **a**. On the STEM-HAADF image, LSCF and GDC regions are denoted by the letters *L* and *G*, respectively.

regions different from those where Ce and Gd simultaneously appear, thereby confirming the co-existence of the two different phases. From these results, it is determined that the dark-contrast nanostripes observed on the STEM-HAADF image are attributed to the LSCF phase, whereas the bright-contrast ones are attributed to the GDC phase (the heavier elements, viz. Gd and Ce, lead to more electron scattering at higher angles). The elemental distribution depicted with Fe and Ce elements overlaid (bottom right) further confirms the arrangement of the LSCF and GDC phases within the nanocomposite film.

To further confirm the crystalline phases comprising the nanocomposite, additional S/TEM and selected area electron diffraction (SAED) patterns were obtained as shown in Fig. 2. Figure 2a highlights a representative granular domain showing nanostripes oriented at an angle to the substrate normal, and Fig. 2b shows the corresponding SAED pattern obtained from the region denoted by a circle in Fig. 2a. Here, we can identify strong diffraction spots indexed to both LSCF and GDC, indicating the highly crystalline nature of the phases exhibiting a quasi-epitaxial relationship. Moreover, shown in Fig. 2c is a representative STEM-HAADF image taken from the same area confirming that the LSCF and GDC phases have formed coherent, quasi-epitaxial interfaces resulting in a remarkably high interfacial density. From additional structural evaluations using X-ray diffraction (XRD), the co-existence of LSCF and GDC phases in the nanocomposite is further corroborated by the presence of peaks attributed to either phase in the characteristic diffraction pattern. However, the peaks attributed to the LSCF phase are relatively weaker

compared to those of the GDC phase, possibly due to the nature of the highly dispersed and discontinuous nanometer-sized grains of LSCF in the nanocomposite structure, in contrast to the well-defined diffraction planes normally present in single-phase LSCF thin films. Typical plan-view SEM images showed that the LSCF-GDC nanocomposite is highly dense with characteristic granular features which appear to be dependent on the specific crystalline orientation of grains comprising the underlying GDC substrate. Additional data are presented in the Supplementary Information (Supplementary Fig. 1: XRD data, Supplementary Fig. 2: SEM).

**Nanoporous LSC layer**. For an oxygen reduction reaction to occur on cathode surfaces, it is essential that the cathode structure is sufficiently porous to fulfill the gas permeability requirement. One particular advantage of PLD is the ease with which the resultant microstructure of the film can be tuned by simply selecting the appropriate deposition parameters, for instance by selecting a parameter region with high oxygen partial pressure and low deposition temperature to obtain highly porous structures[28,29]. For this purpose, LSC thin films ~1 μm thick are subsequently deposited on top of the dense LSCF-GDC nanocomposite layer, by selecting PLD conditions specifically to obtain nanoporous microstructures. To achieve better cathode performance, here we have selected the material LSC due to its inherently higher oxygen surface exchange property as compared to LSCF[30,31].

Typical SEM images depicting the surface and cross-sectional microstructures of the as-grown LSC thin films are shown in Fig. 3a–c. Unlike the LSCF-GDC nanocomposite, which typically exhibits a dense and compact structure, the LSC thin films are comprised of nanoporous agglomerates separated by open pores or nanochannels. As the LSC thin films were deposited at room temperature, the nanostructures would invariably change due to grain coarsening when subjected to higher temperatures; nevertheless, since the as-grown film microstructure is comprised of nanocolumns which are mostly isolated from each other, this strategy seems effective in preventing the excessive grain sintering and densification which would otherwise occur for densely packed nanoscaled structures in direct contact[18]. This is evidenced by the representative cross-sectional SEM images in Fig. 3d, e showing the LSC thin film after an annealing treatment conducted at 700 °C in the air for 1 h, and the assembled cathode layer comprising LSC thin film on LSCF-GDC nanocomposite prepared on GDC electrolyte (likewise heated up to 700 °C in the air) in Fig. 3f. Here, it can be observed that the LSC thin-film nanostructure evolved into clustered nanopillars comprised of grains ranging in 50–100 nm in size, a typical grain coarsening behavior due to sintering induced at high temperatures[21]. Nevertheless, despite the grain coarsening, the LSC thin film still exhibits high porosity for gas permeability as well as high surface area, which are ideal properties for facilitating oxygen reduction reaction on the cathode surfaces. Additionally, the LSC thin film exhibits good adhesion at the interface with the nanometer-sized grains of the underlying LSCF-GDC nanocomposite layer.

**ASR evaluation**. Electrochemical impedance spectroscopy (EIS) measurements were performed at 500–700 °C and the ASR values for symmetrical cells on GDC electrolytes are extracted from the impedance spectra. A summary of the various configurations of the symmetrical cells evaluated in this study is shown in Supplementary Table 1. For all symmetrical cell samples, the thin-film cathodes were contacted by unsintered LSC paste which served as the current collector. In terms of microstructure, the unsintered LSC paste used in this study resembles those in related studies[32–34] which utilized finely grained in-situ activated LSC

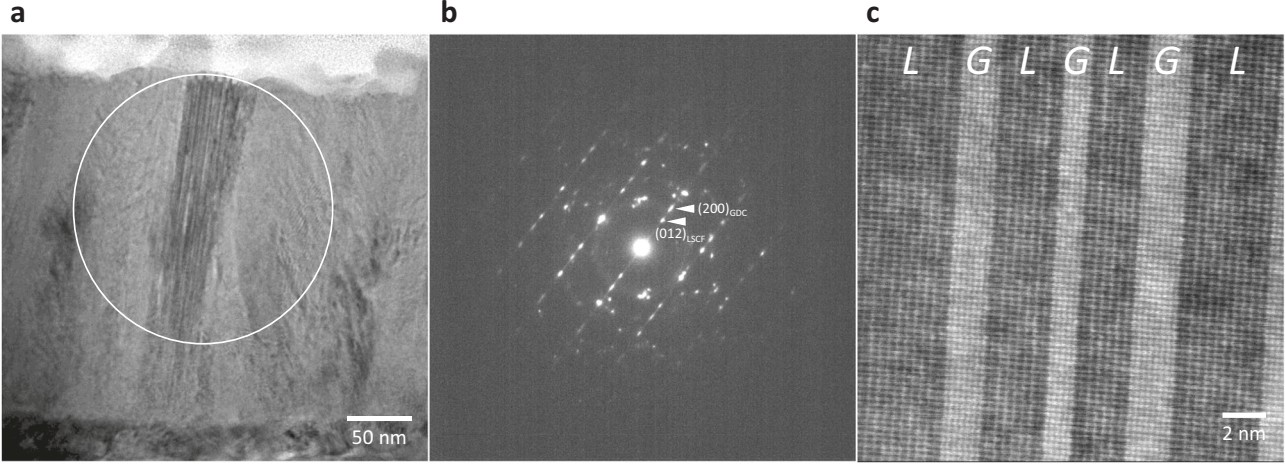

**Fig. 2 Phase identification of the LSCF-GDC nanocomposite film. a** Low-magnification cross-sectional TEM image depicting granular domains containing nanostripes. **b** Selected area electron diffraction (SAED) pattern of the region denoted by a circle in **a**. **c** STEM-HAADF lattice image of the LSCF-GDC nanocomposite film, showing the coherent, quasi-epitaxial interfaces between the LSCF and GDC phases, denoted as *L* and *G* on the image, respectively.

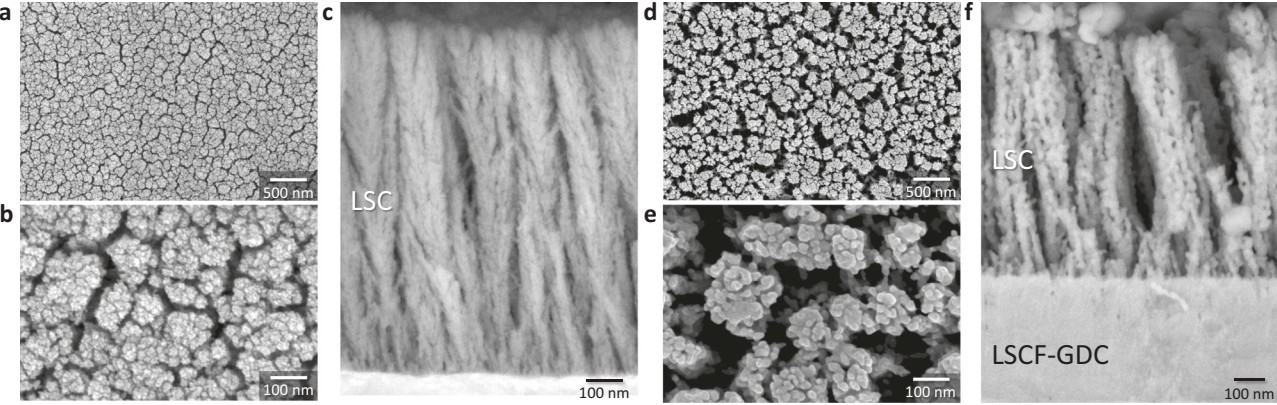

**Fig. 3 Microstructural evaluation of the nanoporous LSC thin film. a** Representative SEM images depicting a low-magnification view of the surface of the as-grown nanoporous LSC thin film, **b** corresponding high-magnification view of the surface, and **c** details of its cross-sectional microstructure. The typical microstructure is comprised of vertically aligned agglomerates separated by nanochannels or nanoporosities. **d, e** Corresponding SEM images depicting the surface of the nanoporous LSC thin film after annealing at 700 °C in the air for 1 h. **f** SEM showing the cross-section of the assembled nanoengineered cathode layer comprising the nanoporous LSC thin film on the LSCF-GDC nanocomposite.

paste as cathode, and may therefore be expected to contribute to the electrochemical activity of the cell. To understand the effect of the unsintered LSC paste, a symmetrical cell containing only this electrode (Sample 1) is also evaluated for comparison. To elucidate the effect on the performance of each layer in the nanoengineered cathode, viz., LSCF-GDC nanocomposite and nanoporous LSC, additional samples having various layer combinations were prepared on GDC electrolytes (refer to Supplementary Table 1): Sample 2 has nanoporous LSC only, Sample 3 has LSCF-GDC nanocomposite layer only, and Sample 4 contains both LSCF-GDC nanocomposite layer and nanoporous LSC. For comparison, Sample 5 is a LSCF thin film prepared using the same deposition conditions as the LSCF-GDC nanocomposite, and further combined with nanoporous LSC.

The Arrhenius plot of the ASR results for the samples is shown in Fig. 4. At 500 °C, the unsintered LSC paste (Sample 1) exhibits a relatively higher ASR value compared to the samples containing thin-film cathodes. In this intermediate temperature range, the ASR performance is considered to be dominated by oxygen exchange kinetics[35,36], and given that the contribution to the electrochemical activity by the unsintered LSC paste may be considered constant among the samples similarly contacted by

this material, this suggests that the improvement in performance observed for the samples containing the thin-film cathodes can therefore be attributed to the superior oxygen exchange properties of these layers. Additional improvement may also be expected due to more intimate contact of the thin films at the interface with the GDC electrolyte as compared to the case of unsintered LSC paste only. On the other hand, at 600–700 °C, with the exception of Sample 4 (LSCF-GDC nanocomposite + nanoporous LSC), the performance of the unsintered LSC paste is comparable to the samples containing the nanoporous LSC (Sample 2 and Sample 5). From these results, it appears that in this temperature range the performance of the cells having the nanoporous LSC layers becomes indistinguishable from that of the unsintered LSC paste alone. One plausible explanation is that the nanoporous LSC interacts with the unsintered LSC paste as the cells are heated, which is quite expected given that this layer was prepared at room temperature and would therefore invariably change upon heating. This indicates a possible limitation on the utilization of in-situ activated cathode materials such as LSC as a current collector for electrochemical evaluation of similarly structured nanoporous cathodes at high temperatures; in this regard, alternative materials should thus be explored.

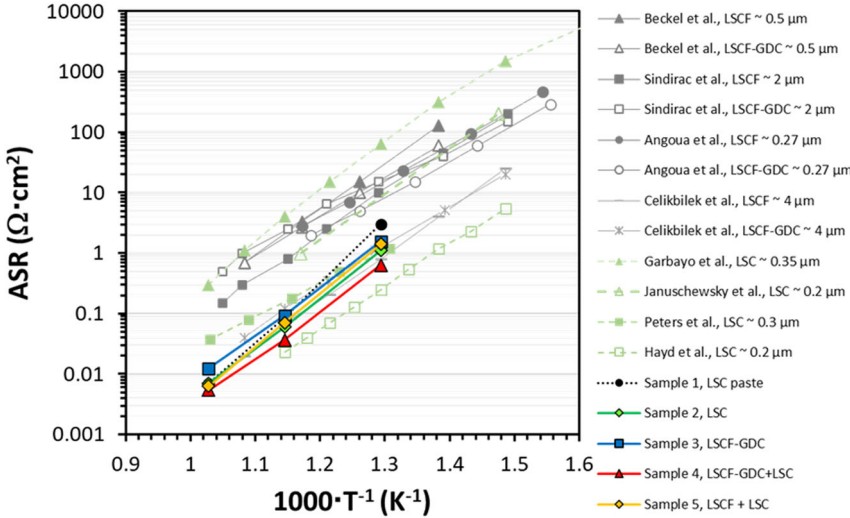

**Fig. 4 Arrhenius plot of the area-specific resistance (ASR).** ASR of the symmetrical cells having various configurations (refer to Supplementary Table 1) in this study are compared with values for LSC, LSCF, and LSCF-GDC nanocomposite cathodes, presented in literature[16,37–39,52–55].

Nevertheless, these results indicate that introduction of nanostructured thin-film cathodes such as the ones in this study could lead to further improvement in the ASR values more especially at lower temperatures.

As shown in Fig. 4, Sample 3 (LSCF-GDC nanocomposite only) exhibits slightly higher ASR values than Sample 2 (nanoporous LSC only), which may be reasonably accounted for by its dense microstructure characterized by nanometer-sized domains and thus fewer electrode-gas phase interfaces and less intimate contact with the current collector, in contrast to that of nanoporous LSC layer. On the other hand, among the samples evaluated, Sample 4 having the combined nanoengineered cathodes (LSCF-GDC nanocomposite + nanoporous LSC) exhibits the lowest ASR values, indicating that the effective combination of the two types of layers results in superior cathode performance.

The important role of the LSCF-GDC nanocomposite as a transition layer is further confirmed from a comparison of the performance of Sample 4 with Sample 5. Here, even though both samples employ the same nanoporous LSC top layer, the utilization of the LSCF-GDC nanocomposite for Sample 4 resulted in about 50% lower ASR value (~36 mΩ cm²) compared to the case of LSCF for Sample 5 (~70 mΩ cm²) at 600 °C. This proves the superiority of utilizing a layer having a structure like the LSCF-GDC nanocomposite as compared to that of a single-phase LSCF thin film as a transition layer at the interface with the GDC electrolyte. This is ascribed to the presence of a high interfacial density which boosts the oxygen diffusion and transfer across the electrode/electrolyte interface.

The electrochemical performances of the samples in this study are further compared against the reported values in the literature for several LSC, LSCF, and LSCF-GDC composite cathodes prepared by other techniques such as spray pyrolysis[37], polymeric precursor[38], and electrostatic spray deposition[39]. Depending on the method adopted for the thin film preparation as well as film thickness, the addition of the GDC phase to the LSCF phase in the cathode structure may either have a positive or negative effect, possibly due to the combined effect of "heterogeneities in porosity within the film thickness and percolation of the ionically conducting phase[39]". Using PLD to prepare the LSCF-GDC nanocomposite (Sample 3) already results to a significant improvement in ASR values compared to those prepared by other techniques, however, further reductions in ASR values were

achieved by incorporating the nanoporous LSC as well (Sample 4). The lowest ASR values obtained for the cells in this study are record-low values for LSCF-type electrodes and are competitive to the highest values reported so far for LSC-type electrodes. Refinement of the nanostructures by controlling the overall morphology and porosities is expected to further improve these values.

**Anode-supported cell fabrication and characterization.** Next, we demonstrate the feasibility of integrating the nanoengineered cathodes on practical cells, namely anode-supported cells prepared using conventional methods. Figure 5a shows the schematic illustration of the cell architecture developed in this study, depicting a Ni-YSZ ($Zr_{0.85}Y_{0.15}O_{1.93}$, 8YSZ) anode-supported cell. The Ni-YSZ substrate has a high porosity of ~50% to reduce gas diffusion resistance. To achieve a pinhole-free, dense YSZ electrolyte layer on Ni-YSZ, which has a typically irregular and porous microstructure, an anode functional layer (AFL) of the same composition as the Ni-YSZ substrate but of smaller particle sizes was designed and fabricated (Fig. 5b). The AFL was prepared using NiO-YSZ nanocomposite particles synthesized via spray pyrolysis for the raw powder material. Due to the low sinterability of the nanocomposite particles obtained through the spray pyrolysis method, the fine AFL structure could be maintained even after high-temperature co-sintering. The preparation of both AFL and YSZ electrolyte layers (~2 μm) follows typical processing routes employed for the manufacturing of practical SOFC cells, however, for our purposes, it is of crucial importance that the GDC interlayer is likewise dense and pore-free, a goal which would be difficult to achieve using conventional screen-printing methods[10,40]. Studies have shown that relatively dense GDC interlayers can be prepared by adopting very high sintering temperatures, which has the disadvantage of inducing chemical interdiffusion between GDC and YSZ, leading to the formation of solid solutions with lowered conductivities[41]. To circumvent this issue, a dense GDC interlayer with a thickness of ~2 μm is prepared using PLD at intermediate temperatures without post-growth sintering. Comparison of the dense GDC interlayers with conventional porous structures and their correlation with ohmic resistance in anode-supported cells is outside the scope of the present study and will be reported in a separate publication. In addition to serving as an effective interfacial barrier, the GDC interlayer simultaneously provides a suitably

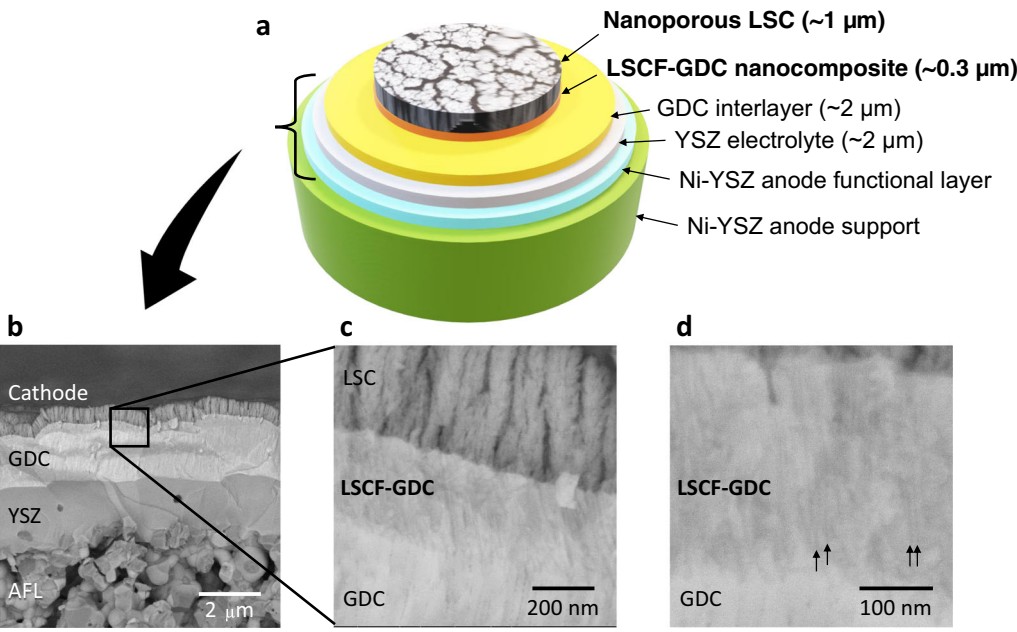

**Fig. 5 Nanoengineered cell architecture and microstructural evaluation of the fabricated cell. a** Schematic illustration of the nanoengineered cell architecture developed for the anode-supported cell. The anode support, AFL, and electrolyte are fabricated using conventional wet processes such as extrusion and screen-printing methods, whereas the GDC interlayer, LSCF-GDC nanocomposite layer, and nanoporous LSC thin film are all deposited using PLD. **b** Cross-sectional SEM image of the as-grown fabricated cell showing the details of the AFL, YSZ electrolyte, GDC interlayer, LSCF-GDC nanocomposite layer, and nanoporous LSC thin film. **c** High-magnification cross-sectional SEM image (the zoomed area is depicted by a square on **b**) showing the detailed microstructure of the GDC interlayer, LSCF-GDC nanocomposite, and nanoporous LSC thin film. **d** High-magnification cross-sectional SEM image showing details of the LSCF-GDC nanocomposite and part of the GDC interlayer as prepared on anode-supported cell. Black arrows indicate dark-contrast stripes pertaining to the LSCF phase in the nanocomposite.

dense and homogeneous surface required for the subsequent deposition of the LSCF-GDC nanocomposite layer. As evidenced by Fig. 5c, the resulting dense GDC interlayer fulfills these requirements, which then enables the successful deposition of the succeeding layers. The nanoengineered cathodes were prepared using identical conditions as those employed for the symmetrical cells on GDC electrolytes, although further optimization of deposition parameters for anode-supported cells may still be explored in future studies. Finally, Fig. 5d depicts a representative high-magnification SEM image acquired in backscattered electron mode, showing the distribution of the LSCF and GDC phases which can be distinguished by the alternately bright and dark nanostripes (indicated by arrows), thereby confirming that the unique self-assembled microstructure of the LSCF-GDC nanocomposite has been successfully replicated on the anode-supported cell. Furthermore, all the PLD-deposited layers (viz. GDC interlayer, LSCF-GDC nanocomposite, and nanoporous LSC thin film) exhibit conformal growth on top of the YSZ electrolyte, which ensures uniform coverage despite the intrinsic unevenness of its surfaces. Detailed microstructural characterization of the different layers in the cell are further presented in the cross-sectional SEM images shown in Supplementary Figs. 3 and 4, which show the completed cell right after the PLD deposition of the various layers.

Analogous to the symmetrical cells presented earlier, to elucidate the effect on the electrochemical performance of the individual cathode layers on the anode-supported cell, *viz.* nanoporous LSC and LSCF-GDC nanocomposite, we evaluated three cells with various configurations. Sample 6 has LSCF-GDC nanocomposite layer only, Sample 7 has nanoporous LSC only, and Sample 8 has both layers (refer to Supplementary Table 1). The cells have an active electrode area of 0.785 $cm^2$ and were

tested under a supply of 3% humidified $H_2$ as fuel to the anode and of dry air as an oxidant to the cathode. Here, similar to that employed for symmetrical cells, unsintered LSC paste was used as the current collector. The current–voltage ($I$–$V$) and current–power ($I$–$P$) curves of the anode-supported cells are shown in Fig. 6. The performance was measured at temperatures ranging from 600 to 700 °C. High OCV (open-circuit voltage) values of ~1.1 V, close to the theoretical value of 1.120 V were obtained at all temperatures. The overall trend shows that employing the LSCF-GDC nanocomposite layer alone (Sample 6) is insufficient to enhance the maximum power density compared to the one combined with nanoporous LSC and moreover, it shows the lowest performance among the samples at all temperatures evaluated. The $I$–$V$ dependence for this sample also shows a more prominent positive curvature at high current densities, indicating gas diffusion polarization[42]. This may be ascribed to the relatively dense microstructure of this layer where gas diffusion is expected to be limited. On the other hand, the nanoporous LSC layer alone (Sample 7) shows better performance than the LSCF-GDC nanocomposite alone at all temperatures; however, analogous to the results obtained for symmetrical cells, we note that it is actually the combination of both layers (Sample 8) that yields the best performance and highest maximum power output among the cells at any temperature. These results indicate that the optimum cathode performance can thus be obtained by effectively combining both nanoporous LSC and LSCF-GDC nanocomposite as cathode layers in the cell architecture.

With the goal of overcoming limitations to properly evaluate the performance of the nanoengineered cell architecture, the effect of further modifications in the measurement procedures was investigated. As the results showed earlier revealed, the use of

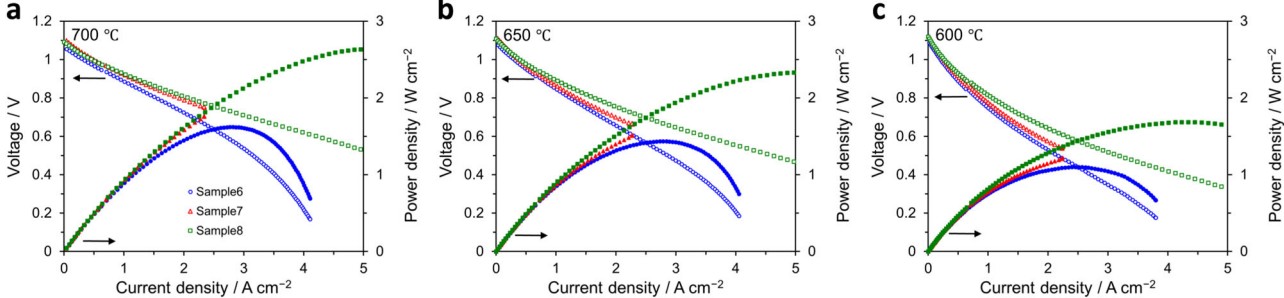

**Fig. 6 Comparison of electrochemical performance of anode-supported cells.** Current–voltage (*I–V*) and current–power (*I–P*) curves were evaluated at **a** 700 °C, **b** 650 °C, and **c** 600 °C for different anode-supported cell configurations, namely Sample 6 (LSCF-GDC, blue circles), Sample 7 (LSC, red triangles), and Sample 8 (LSCF-GDC + LSC, green squares). The active electrode area was 0.785 cm² and the current collector was unsintered LSC paste for all cells.

unsintered LSC paste as a current collector could have limited the evaluation of cell performance at higher temperatures, which then leads to the question of whether utilizing a different material for current collection would allow a more accurate evaluation of the cell. To gain a better understanding of this effect, cells of the same architecture as Sample 8 were measured using Pt paste as a current collector. The results are then compared as shown in Fig. 7.

The change from LSC paste (left-hand side) to Pt paste (right-hand side) as current collector enabled the attainment of better performance at 700 °C, even though at lower temperatures of 650 and 600 °C the performances of the two cells are quite comparable. This does not necessarily imply that the Pt paste somehow interacted with the thin-film cathodes, leading to an apparent improvement in performance; the SEM image shown in Fig. 7f depicts the cross-section of the cell after testing and confirms that the Pt did not infiltrate the nanoporous LSC layer. Unlike the LSC paste, the Pt paste could be detached easily after testing, with very few residues (indicated by an arrow) located only on the surface of the nanoporous LSC, indicating no chemical interaction which could possibly affect the cathode performance. Therefore, we believe that for this case utilizing the Pt paste most likely enables a more accurate evaluation of the thin-film cathodes at higher operating temperatures. To reiterate, the main implication here is that the choice of material to be used for the current collection is and should be an important consideration for the evaluation of the performance of thin-film cathodes[26,27]. In practical applications, though Pt will not likely be the material of choice due to its high cost, we believe that these results still provide useful insights into designing current collectors appropriate for use with thin-film cathodes in future studies.

To check the reproducibility of the performance shown in Fig. 7, a cell with the same configuration (Sample 9 in Supplementary Table 1) was prepared and measured on a different test rig equipped with a higher maximum current load. For this test rig, the active area was limited to 0.283 cm²; however, as the films were prepared using the same deposition conditions, a reduction in the cathode area is not expected to have a direct influence on the performance.

Figure 8 shows the electrochemical performance of the optimized cell having both nanoporous LSC and LSCF-GDC nanocomposite layers, with Pt paste utilized as a current collector. Similar to the evaluation for the previous samples, this sample was evaluated at different temperatures under a supply of 3% humidified H₂ as fuel to the anode and of dry air as an oxidant to the cathode. Although the active electrode area was reduced for this measurement, we obtained almost similar performance in

terms of area-specific ohmic resistance and polarization resistance values based on comparison with the previous cell evaluated with Pt paste (Fig. 7b, d), confirming the reproducibility of the quality of the cells irrespective of the measurement system.

As shown by the *I–V* characteristics in Fig. 8a, the measured OCV at 600 °C is ~1.1 V, and current densities of ~2.2 and ~4.7 A/cm² under an operating voltage of 0.7 V at 650 and 700 °C, respectively, were successfully obtained. This corresponds to power densities of ~1.5 and ~3.3 W/cm², and are competitive to the highest reported values in the literature for anode-supported cells[43] or metal-supported cells[32] utilizing high-performance LSC cathodes. Figure 8b shows the impedance plots at 0.75 V and Fig. 8c shows the Arrhenius plot of the ASR values as determined from the impedance spectra. This cell exhibits a low ohmic resistance $R_{ohm}$ of 0.026 Ω cm² and polarization resistance $R_p$ of 0.030 Ω cm² under an operating condition of 0.75 V, which are attributed to the utilization of a thin YSZ electrolyte and dense GDC interlayer, and high oxygen exchange property ascribed to the nanoengineered cathode layers, respectively. Significant reduction of these parameters enabled the attainment of superior power output for the anode-supported cells. We also evaluated the electrochemical performance of another similarly designed cell and separately measured as shown in Supplementary Fig. 5. The results showed good agreement with those shown in Fig. 8, indicating good reproducibility of our adopted nanoengineered architecture for anode-supported cells. In addition, it may be noticed that both measured *I–V* curves exhibit a characteristic non-linear behavior that suggests either a possible activation with current density or an increase in cell temperature due to self-heating. The phenomenon of self-heating at high current densities has been reported for large cells utilizing conventional cathodes[32]. However, comparison of the ohmic area-specific resistance values of the cells in this study obtained at 0.75 V and OCV (Supplementary Fig. 5b, c, respectively) at various temperatures do not show any significant differences, which suggest that self-heating effects are unlikely to have caused this behavior. Instead, this anomalous behavior appears to be directly correlated to the complexity of the nanostructures developed in this study, which could result in unique properties highly different from those typically exhibited by conventional cathodes. Our ongoing studies seek to elucidate the mechanisms governing this behavior through detailed analyses of the nanoengineered cathodes using advanced characterization tools and will be reported in a future publication.

Table 1 shows a summary of electrochemical performance reported in the literature for selected anode-supported cells utilizing various cathodes. Here the current density for an operating voltage of 0.7 V is taken as the measure of

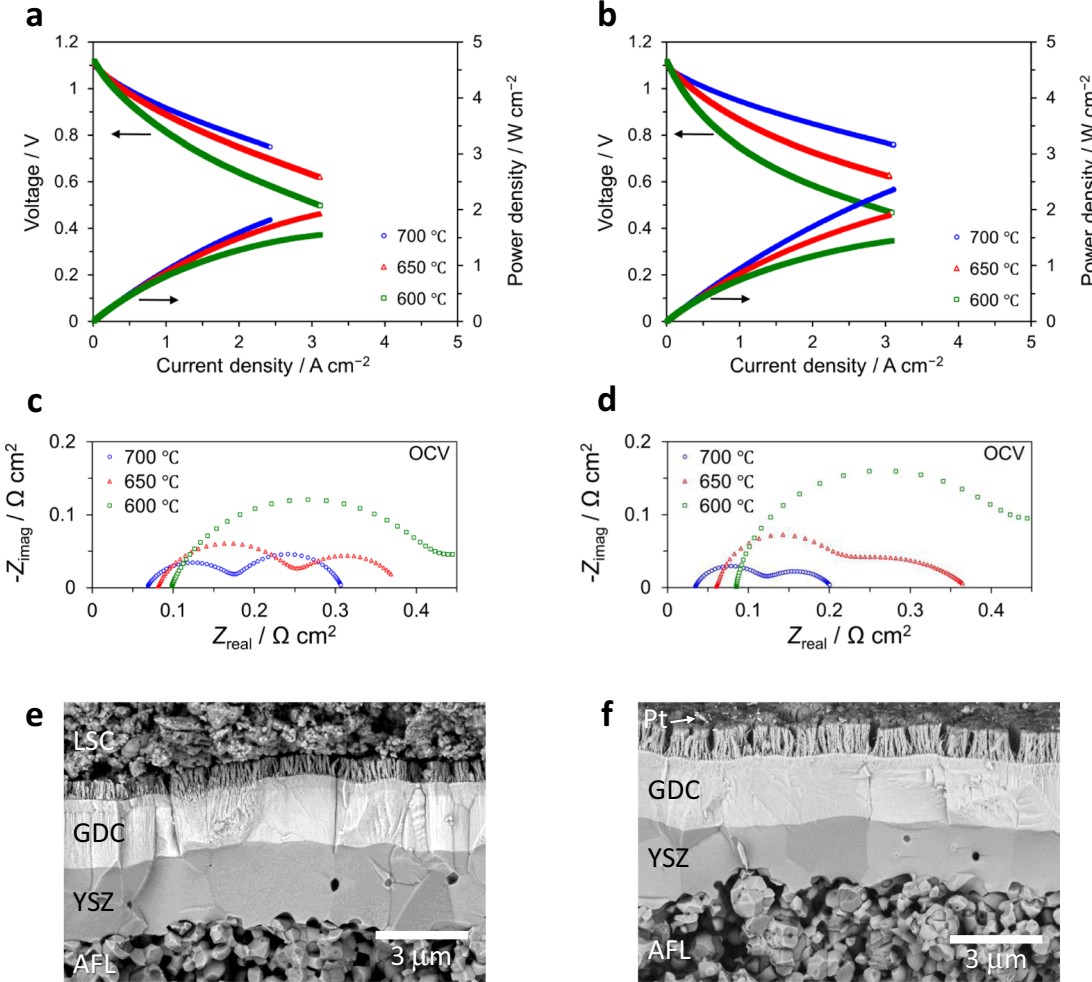

**Fig. 7 Comparison of electrochemical performance and microstructure of anode-supported cells. a** Current–voltage (*I–V*) and current–power (*I–P*) curves evaluated at various temperatures (700 °C: blue circles, 650 °C: red triangles, 600 °C: green squares) of the anode-supported cells utilizing nanoengineered cathode layers comprised of nanoporous LSC and LSCF-GDC nanocomposite (Sample 8 configuration, see Supplementary Table 1) and tested using different current collectors. LSC paste was utilized for the cell on the left-hand side, whereas Pt paste was used for the one on the right-hand side. The current load is limited to 3 A/cm². The performance at 700 °C for the cell tested with LSC paste is inferior compared to the one where Pt paste was used; however, their performances are comparable at lower temperatures (650 and 600 °C). These results suggest that the material used for the current collection is an important consideration when evaluating the electrochemical performance. The active electrode area for both cells is 0.785 cm². **c**, **d** Corresponding impedance spectra at OCV and various temperatures. **e**, **f** Representative cross-sectional SEM images showing the cells after testing using LSC paste (**e**) and Pt paste (**f**) as current collectors.

electrochemical performance. The performance of cells utilizing the nanoengineered cathode layers developed in this study is about a factor of three higher than the top-performing LSCF-GDC composite cathodes on anode-supported cells reported in the literature (cf. performance at 700 °C, Park et al.[44]) and outperforms that recorded to date for a thin-film composite electrode by about 20% (cf. SSC-SDC composites on LSGM, Kang et al.[20]) at the operating temperature of 700 °C. The superior performance of our cell is attributed to the combination of the following reasons: first and foremost, the enhanced cathode performance resulting in very low ASR as facilitated by the combined nanoporous LSC thin film and highly ordered self-assembly of LSCF and GDC in the nanocomposite layer, and next, its successful integration via the nanoengineered cell architecture adopted in the fabrication of the cell, which among others employs a relatively thin YSZ electrolyte and a dense GDC interlayer which effectively eliminates the need for post-growth sintering. In this configuration, we can therefore harness the superior electrochemical performance of the thin-film

cathodes and achieve high power densities which could not be easily attained with conventional, micrometer-scaled cathode structures. For industrial-scale applications, large-area PLD or sputtering systems with capabilities of depositing on substrates with sizes of 8 inches in diameter or more are already commercially available[45] and can be utilized to fabricate similar nanoengineered cathodes on several large-area substrates in batch processes. Due to the potential of achieving significantly higher power densities over relatively smaller areas, it would also be possible to develop more compact devices by utilizing nanoengineered thin-film cathodes. Moreover, the proposed cell architecture can be readily adapted to smaller-scale micro-SOFC devices, in combination with metal substrates and other low-cost, practical materials. The results presented here mark an important first step in the development of similarly designed innovative materials for high-performance SOFC devices.

Lastly, as a preliminary test to evaluate the cell performance over time, the voltage was measured at a constant current of 1 A/

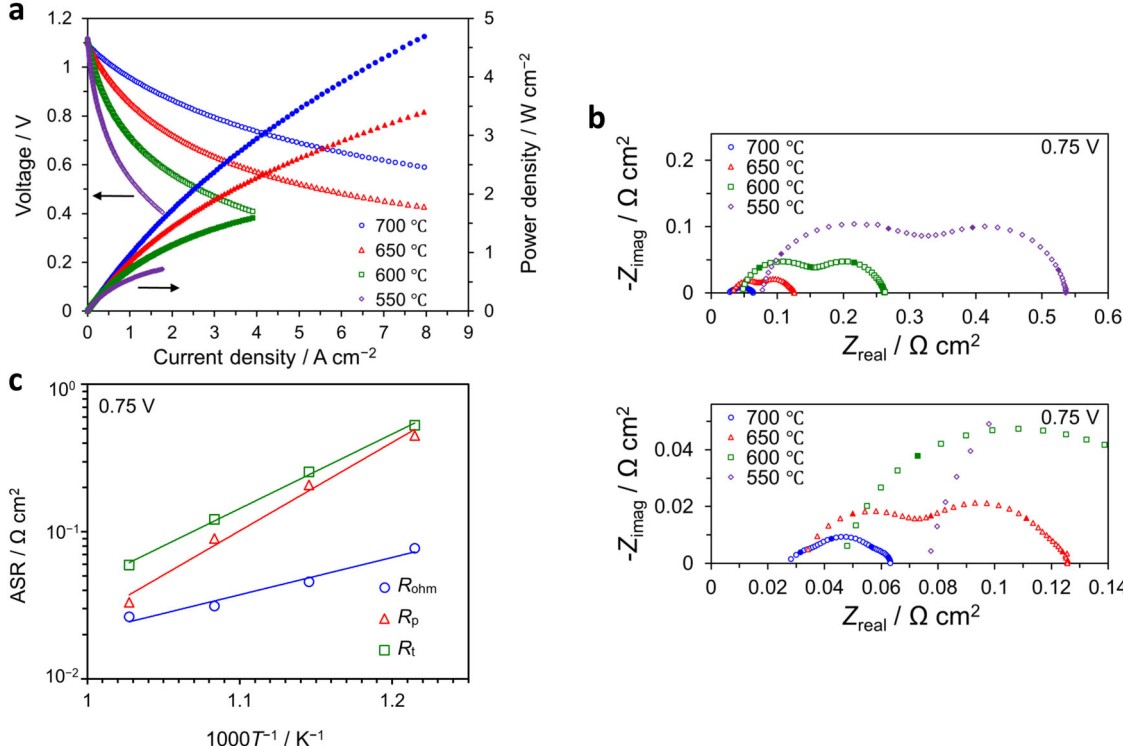

**Fig. 8 Electrochemical performance of anode-supported cell.** This cell, utilizing a nanoengineered cathode layer comprising nanoporous LSC and LSCF-GDC nanocomposite, was evaluated using humidified (3% $H_2O$) $H_2$ as fuel and dry air as oxidant at various temperatures (700 °C: blue circles, 650 °C: red triangles, 600 °C: green squares, 550 °C: purple diamonds). **a** Current–voltage (I–V) and current–power (I–P) curves evaluated at various temperatures. **b** Impedance spectra for the anode-supported cell at an operating condition of 0.75 V. **c** ASRs for the anode-supported cell as determined from impedance spectra ($R_p$: polarization resistance, $R_{ohm}$: ohmic resistance, $R_t$ = total resistance).

**Table 1 Electrochemical performance of selected anode-supported cells.**

| Component materials | | | | $j_{0.7\ V}$ [A/cm²] | Refs. |
|---|---|---|---|---|---|
| **Cathode** | **Interlayer** | **Electrolyte** | **Anode** | | |
| LSCF-GDC | GDC | YSZ, ~2 µm | Ni-YSZ | 4.72 (700 °C) 2.21 (650 °C) | This study |
| LSCF-GDC[a] | – | YSZ, ~2.5 µm | Ni-YSZ | 1.70 (700 °C) | Park et al.[44] |
| LSCF-GDC | GDC | YSZ, ~5 µm | Ni-YSZ | 0.87 (650 °C) | Sumi et al.[47] |
| LSCF-GDC | – | GDC, ~49 µm | Ni-GDC | 0.63 (650 °C) | Leng et al.[48] |
| LSCF-GDC | – | GDC, ~14 µm | Ni-GDC | 0.27 (650 °C) | Fu et al.[49] |
| LSCF-GDC | – | GDC, ~25 µm | Ni-GDC | 0.58 (650 °C) | dos Santos-Gomez et al.[50] |
| SSC-SDC[b] | GDC | YSZ, ~6 µm | Ni-YSZ | 1.60 (700 °C) | Shimada et al.[51] |
| SSC-SDC[b] | – | LSGM, ~5 µm | Ni-Fe | 3.90 (700 °C) | Kang et al.[20] |

$j_{0.7\ V}$ current density at 0.7 V.
[a]$PrO_x$-infiltrated
[b]SSC-SDC, Sr-doped $SmCoO_3$ (SSC) and Sm-doped $CeO_2$ (SDC)

cm² at 700 °C for more than 250 h. As the results in Fig. 9a show, although there was a drastic drop in voltage within the first ten hours, the cell performance is quite stable at ~0.8 V throughout the time range evaluated. Figure 9b shows the I–V and I–P curves evaluated at 700 °C at the end of the durability test, which indicates that effective measures must be further developed in order to improve the stability in performance. Systematic investigation of the various components comprising the entire cell may provide important clues towards understanding the origin of the initially rapid degradation and how such behavior can be mitigated in long-term operation. It is hoped that our ongoing studies will help to identify the critical factors affecting the cell durability and identify specific strategies such as operating at relatively lower temperatures, optimization of microstructures

of the various components, and fabrication processes, among others.

## Discussion

High-performance, nanoengineered cathodes comprising nanoporous LSC thin films in conjunction with self-assembled LSCF-GDC nanocomposites were developed. Gas permeability of the LSC thin film is ensured by its highly nanoporous structure having a high surface area even with grain coarsening which inevitably occurs upon exposure to high operating temperatures. On the other hand, LSCF and GDC self-assemble into a highly ordered, nanocomposite layer with superior interfacial stability via the coherent, quasi-epitaxial interfaces between the LSCF and

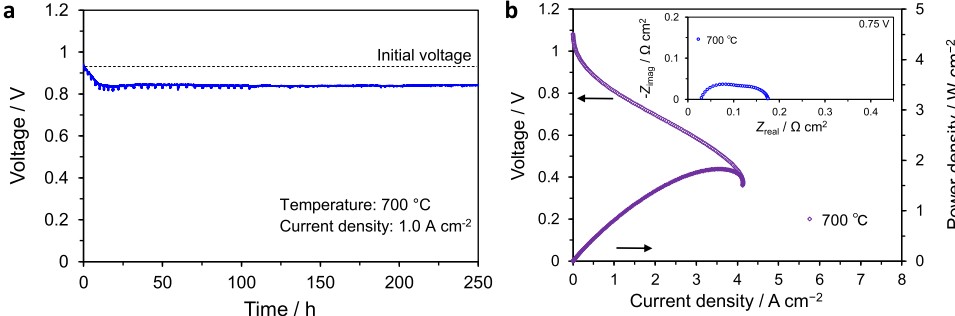

**Fig. 9 Evaluation of cell performance with time. a** Time-dependence of the voltage at an applied current density of 1 A/cm² at 700 °C. **b** Current–voltage (*I–V*) and current–power (*I–P*) curves evaluated at 700 °C at the end of the durability test. Inset shows the final impedance spectra obtained at 0.75 V and 700 °C.

GDC phases. The combination of these two layers in the cathode structure resulted in superior cathode performance with very low ASR values competitive to the highest values reported in the literature. The nanoengineered cathode layers were successfully integrated into conventional anode-supported cells, whereby the utilization of a relatively thin YSZ electrolyte in combination with a dense GDC interlayer, facilitates significant reduction in the cell's ohmic resistance complementary to the superior cathode performance. Consequently, anode-supported cells utilizing the nanoengineered cathodes demonstrate superior cell performance in terms of high current densities at 0.7 V achieving ~2.2 and ~4.7 A/cm² at 650 °C and 700 °C, representing a significant improvement of performance compared to hitherto reported cells utilizing thin-film cathodes. Further work is ongoing to fine-tune the cell architecture, optimize the fabrication process, and elucidate the factors affecting long-term performance stability. These advancements are expected to pave the way towards applications in next-generation SOFCs.

## Methods

**Film growth**. The LSCF-GDC nanocomposite thin films were grown via a simple one-step process using a PLD system (NanoPLD, PVD Products). A polycrystalline target consisting of LSCF and GDC (10 mol% Gd₂O₃) sectors with an areal ratio of 3:1 was ablated by a KrF excimer laser (248 nm wavelength, COMPexPro 102, Coherent) with laser energy of 200 mJ and repetition rate of 10 Hz. The samples were heated at 750 °C at 5 Pa oxygen pressure and after film deposition were chamber-cooled to room temperature under the same atmosphere. GDC thin films as interlayers were prepared using the same deposition conditions as the LSCF-GDC nanocomposite thin films, with the desired thickness obtained by increasing the number of laser pulses. LSC thin films were grown using LSC ceramic targets ablated with laser energy of 275 mJ and repetition rate of 20 Hz at room temperature and 13.3 Pa oxygen pressure, and the number of laser pulses was adjusted to achieve a final thickness of ~1 μm. Identical deposition conditions were adopted for the preparation of cathode layers on both symmetrical cells and anode-supported cells. Supplementary Table 1 shows the summary of all the samples prepared in this study.

**Characterization**. Microstructural characterization was performed using a field-emission scanning electron microscope (SEM, NovaNanoSEM 450, FEI) in secondary electron and backscattered electron mode, and transmission electron microscope (TEM, Tecnai Osiris, FEI) operated at 200 kV in scanning TEM mode (STEM). Energy-dispersive X-ray spectroscopy (EDX) measurements were performed in the same instrument. Cross-sectional samples for S/TEM characterization were prepared using a dual-beam FIB-SEM (focused ion beam, Scios, FEI) equipped with Ga⁺ ion source, operated with a maximum acceleration voltage of 30 kV and final thinning performed at <3 kV. For further thinning and to reduce the damage from the FIB milling, additional milling was performed using an argon ion milling machine (Gatan PIPS 691 precision ion polishing system) operated at room temperature.

For the electrochemical impedance measurement of symmetrical cells, an unsintered LSC paste consisting of fine particles ranging from hundreds of nm to ~1 μm in sizes was applied to the samples using the screen-printing method, allowed to dry at 100 °C inside a drying oven for 12–24 h, afterward gold mesh was

pressed as a contact. The active electrode area was 0.785 cm². Electrochemical impedance spectra were recorded with a Versastat4 (Princeton Applied Research, USA) frequency response analyzer in a frequency range of 1 MHz to 0.1 Hz from 500 to 700 °C in flowing air (50 ml/min). The impedance data were analyzed and fitted using the Zview software.

**Anode-supported cell fabrication**. NiO powder (Sumitomo Metal Mining Co.), YSZ powder (TZ-8Y, Tosoh) at a weight ratio of 60:40 were mixed with pore former and binder, and then the powder mixture was stirred in a vacuum chamber with the addition of a proper amount of distilled water. After aging the mixture in an ambient atmosphere for 15 h, the mixture was formed into a NiO-YSZ green sheet by an extrusion process where a metal mold (0.7-mm thick, 120-mm wide) was used. The detailed fabrication procedure of the extrusion process has been presented elsewhere[46]. Button-size pieces (32-mm diameter) cut from the NiO-YSZ green sheet were sintered at 1240 °C for 2 h in air. The NiO-YSZ paste for the AFL was prepared by mixing NiO-YSZ nanocomposite particles synthesized by spray pyrolysis with α-terpineol (Kanto Chemical Co.), ethyl cellulose (45 cP, Kishida Chemical Co.), dispersant, and plasticizer. The YSZ paste for the electrolyte layer was also prepared by mixing YSZ powder (TZ-8Y, Tosoh) and the same admixtures as in the NiO-YSZ paste. On the sintered NiO-YSZ pieces, the NiO-YSZ paste and YSZ paste were screen-printed in succession, and then the coated pieces were co-sintered at 1360 °C for 3 h in air, resulting in electrolyte/anode half-cells (~22 mm in diameter) with a NiO-YSZ AFL. To complete the cell, the GDC interlayer, LSCF-GDC nanocomposite, and nanoporous LSC thin films were deposited using the PLD technique with deposition conditions as described above.

**Anode-supported cell performance measurements**. For Samples 6, 7, and 8, current–voltage (*I–V*) and EIS measurements were performed for the anode-supported cells at 600–700 °C using 3% humidified H₂ as fuel at a flow rate of 200 mL/min and ambient air as oxidant at a flow rate of 200 mL/min by sealing with pyrex glass ring. A potentiostat/galvanostat with a frequency response analyzer (VSP-300 with 10 A/5 V booster board, Biologic) was used for these measurements. In EIS, the frequency range was 1 MHz to 0.1 Hz and the amplitude of applied voltage was 10 mV. For the current collection, unsintered LSC paste similar to that utilized for symmetrical cells was painted on the top surfaces of the cathode, and nickel foam contacted the anode. The operating temperature was monitored using a thermocouple located in proximity (~1 mm) to the anode surface. A schematic illustration of the electrochemical test set-up is shown in Supplementary Fig. 6.

For Sample 9 and other samples with a similar configuration, *I–V* and EIS measurements were performed in a different test bench for the anode-supported cells at 550–700 °C using 3% humidified H₂ as fuel at a flow rate of 70 mL/min and ambient air as oxidant at a flow rate of 140 mL/min. A potentiostat/galvanostat with a frequency response analyzer (Autolab PGSTAT302, Metrohm) was used for these measurements. In EIS, the frequency range was 1 MHz to 0.1 Hz and the amplitude of applied voltage was 10 mV. For the current collection, unsintered Pt paste was painted on the top surfaces of both cathode and anode. The operating temperature was monitored using a thermocouple located in proximity (~1 mm) to the cathode surface.

## Data availability

The data that support the findings of this study are available from the authors on reasonable request, see author contributions for specific data sets.

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

## Acknowledgements

This work was supported by the Advanced Technology Consortium for Solid State Energy Conversion (ASEC). We are grateful to K. Ogasawara and M. Sugasawa for assistance in the PLD and EIS measurements, and N. Saito for the S/TEM measurements. We thank T. Horita and Y. Fujishiro (AIST), K. Yamagiwa (NTK), H. Sumi (Morimura SOFC Technology), H. Ohnishi (Osaka Gas), T. Fujimoto (Kyocera), K. Kobayashi (Denso), and Y. Tanaka (Miura) for their helpful comments and discussion of the results.

## Author contributions

K.D.-B., T.I., H.K., and K.Y. conceptualized the study. K.D.-B. designed and supervised the PLD experiments, prepared the TEM specimens by FIB, carried out SEM observations, and wrote the first draft with additional input from H.S. who carried out the anode-supported cell fabrication and electrochemical measurements. T.I. carried out additional electrochemical measurements with input from K.Y. H.K. and K.Y. helped in the interpretation and analysis of the results. All authors discussed the results and further developed the manuscript.

## Competing interests

The authors declare the following competing interests: a patent application covering this work has been filed by the National Institute of Advanced Industrial Science and Technology to the Japan Patent Office that name K.D.-B., T.I., H.K., and K.Y. as inventors (Application Number: PCT/JP2020/027490; status of application: pending), covering the aspect of manuscript pertaining to the nanocomposite LSCF-GDC and LSC thin-film electrodes prepared via PLD. A patent application covering this work has been filed by the National Institute of Advanced Industrial Science and Technology to the Japan Patent Office that name H.S. (Application Number: 2020-111653; status of application: pending) as an inventor, covering the aspect of manuscript pertaining to the Ni-YSZ anode functional layer prepared via spray pyrolysis technique. There are no other competing interests.
