## [Peer Review File · Nature Communications]

Reviewers' Comments:

Reviewer #1:

Remarks to the Author:

In this manuscript, Katherine Develos-Bagarinao et al. from AIST, Japan reported the achievement of ultrahigh power density at reducing temperature (550-700°C) by using a thin-film YSZ based electrolyte (~2 μm) with (~2 μm) GDC buffer layer and nano engineered cathode layer with one thin LSCF+GDC self-assembled layer (~0.3 μm) with ordered structure and a porous nonporous LSC (~1 μm) of cathode, as prepared by PLD method, which was claimed to demonstrate a peak power density of 3.7 and 4.7 W cm⁻² at 650 and 700°C, respectively, which should be the record-breaking performance in the world. I believe the main creativity of this research is the the highly ordered self-assembled nano composite LSCF+GDC interlayer, and the most attractiveness of this research is the the ultrahigh performance, which suggests YSZ could also be used in IT-SOFC with highly attractive low-temperature performance (>1.6 W cm⁻² at 600°C). If the results are reproducible, it may change our thinking of development of IT-SOFC that doped ceria electrolytes and protonic electrolytes that are believed to show higher ionic conductivity at lower temperature are more suitable for IT-SOFCs. However, several critical issues should be explained or clarified before I can suggest the potential publication of this manuscript in Nature Commun.

- 1) As we know, PLD was intensively used to prepare thin film electrolyte or electrode in SOFCs more than 5 years ago, so the concept of use of PLD in SOFC is not new. The main creativity is the preparation of a highly ordered LSCF+GDC dense interlayer. However, very recently, Yufei Song et al. reported the self-assembly concept in the preparation of composite electrode for SOFC (Yufei Song et al. Self-Assembled Triple-Conducting Nanocomposite as a Superior Protonic Ceramic Fuel Cell Cathode, *Joule* 3 (11), 2842-2853) what is the difference between the two self-assembled concepts should be explained. Otherwise, the novelty of this research will be greatly reduced.
- 2) Typically, the effective layer thickness of a cathode in SOFC is around 10 μm, this is the reason that most SOFC cathode is prepared to be more than 10 μm in thickness. However, in this study, the authors just prepared the nonporous LSC layer of 1 μm, even including the LSCF+GDC layer, it is also only 1.3 μm, far less than 10 μm. It is recommended that the authors should increase the cathode thickness to larger to see if any beneficial effect, although the cathodic performance base don't he results are already very good
- 3) One of my major concern is the accuracy of the measurement. For thin film electrolyte, the gas leaking between the anode and cathode chamber is very easily to appear, while the direct non electrochemical oxidation of the mixed H₂ and O₂ will make the cell temperature higher than the furnace temperature, that will make the cell power output higher than the actual value at set temperatures. The authors used 2 μm YSZ electrolyte and 2 μm GDC buffer layer, both of them will contribute to the ohmic resistance of the cell. The maximum power density of a cell can be calculated based on $PPD = OCV \cdot OCV / 4 \cdot (\text{total cell resistance})$. Since the cell OCV is around 1.1 V to achieve a peak power density of 4.7 and 3.7 W cm⁻², it means the cell total resistance should be 0.0644 ohm cm² and 0.0818 ohm cm², respectively at 700 and 650 °C. However, based on EIS in Fig.7, the value at 700°C matched well, but the value at 650°C (0.125 ohm cm²) is much larger. At lower temperature such discrepancy is even more obvious. The authors should provide a reasonable explanation.
- 4) As to the stability/durability issue, although the authors tested for a period of 250°C at 700°C at 1.0 A cm⁻², the cell power output is fairly stable. However, if we compared the results in Fig. 7 and Fig. 8, at the initial test, no obvious concentration polarization was observed at high current density, however, at the end of the stability test, based on the I-V curve, large drop in cell voltage at high current density suggests the appearance of concentration polarization. It implies, the electrode is likely sintered. SEM observation of the tested cell after the durability test should be provided.
- 5) The advances in cathode development is very fast during the past several years, however, the authors' reference is mostly before 2018. I would suggest the update of references by providing more recent publications .

If above comments are well addressed, I would suggest the publication of this manuscript in Nature

Commun.

Reviewer #2:

Remarks to the Author:

This manuscript with title "Nanoengineering of cathode layers for solid oxide fuel cells to achieve superior power densities" presents a very significant break-through in IT-SOFC power density by using nanoengineered cathode electrode layers, namely LSCF-GDC nanocomposite transition layer and LSC nanoporous layer. The data and image quality fulfills the requirements of Nature Communication. Despite the significance of the manuscript, I would request for some clarifications and major revisions especially for justifying the high performance reported by the authors:

1. Page 5 Line 118-121: Can the authors explain how 100 times or more cathode/electrolyte interfacial area is calculated? Which baseline are you comparing to?
2. Page 8 Line 191: Authors mention that "it is essential that the cathode structure is sufficiently porous to fulfill the gas permeability requirement." Similar to nanoporous LSC layer, to my understanding LSCF-GDC is also an electrode. Can the authors explain the reason why the LSCF-GDC nanocomposite layer does not require porosity for gas permeability? Wouldn't it be more beneficial if the LSCF-GDC nanocomposite layer is also porous so that electrochemical reaction occurs even closer to the GDC electrolyte?
3. Page 10 Line 246-248: This is not a fair comparison because Sample 4b does not have GDC. Composite electrode such as LSCF-GDC is known to improve cell performance compared to single phase electrode such as LSCF even with conventional wet deposited layer, due to the improved ionic conductivity. The significance of LSCF-GDC nanocomposite layer fabricated by PLD in comparison to a conventional LSCF-GDC composite layer engineered by conventional methods is not evident here.
4. Page 15 Line 351: Can you please explain in detail here how a positive curvature is correlated with activation polarization? It would be helpful if you can also show dV/dI curve and explain the difference among these three samples.
5. Page 15 Line 351-352: Following the above question, doesn't this dense microstructure of LSCF-GDC nanocomposite layer also exist in Sample 7? Why does Sample 7 not showing a positive curvature? Please explain.
6. Page 15 Line 370: This is one of my biggest questions for this manuscript. Is the significant higher performance due to the smaller active area or is it due to the use of Pt paste? Can the authors provide performance of a cell with 0.785 cm² active area but with Pt paste as current collector, so that we know the practical maximum power density for a single cell with your technology? Also, what is the cell total area? A very small active area to total area ratio is not realistic in the stack due to waste of extra anode materials. Please justify and explain.
7. Page 16 Line 378-381, "We speculate...smaller electrode area". This argument doesn't make sense. The number of structural defects in YSZ and GDC in a unit area should be equal as you are using PLD for fabrication. When increasing the active area of the electrode, the area specific resistance should therefore be equal, theoretically. Again, a sample with the same configuration of Sample 8, but with Pt paste is necessary to separate the contribution of Pt paste and the contribution of smaller active area. Please provide additional data to justify.
8. Page 16 Line 384: "measured open-circuit voltages (OCV) at 700 °C is 1.093 V which is almost equal to the theoretical OCV". This is not correct. With 3% humidified H₂ and dry air, the theoretical OCV of the cell at 700 C should be equal to 1.120 V, which is much higher than 1.093 V. One could

argue that this cell has some leakage which cause local temperature of the cell higher than 700 C, resulting in a high power density value. Please explain.

9. Page 16 Line 395, Figure S8: The cell in Fig. S8 seems to have higher power density at 0.75 V, why don't you report this cell in the main text, instead of Sample 8?

10. Page 18 Table 2: Once again, we need to first justify the contribution of using a small active area. This table of comparison may not be fair, as we don't know the "active area to cell total area ratio" for these cells in literature. One could argue that the high performance reported in this paper is due to a very small active area. Again, for comparison, a reasonable and realistic active area should be used. By doing that, the readers can then better understand the potential of your technology.

11. Page 19 Fig. 8b: In Fig. 8b, can the authors also include the performance of the cell before testing, to be compared with data after testing?

12. Page 23 Line 546-547: Why are the flow rates for fuel gas and air different for Sample 8 from those for Samle 5, 6, 7?

Reviewer #3:

Remarks to the Author:

The manuscript describes an innovative approach to produce functional layers of a solid oxide fuel cell (SOFC) using Pulsed Laser Deposition (PLD). While PLD is (at least in the academic world) routinely used to produce SOFCs, the authors present an interesting approach in fabricating a dual-phase interfacial layer between the electrolyte and the air electrode, and provide evidence that this layer is beneficial for cell performance.

Overall, the quality of the manuscript is very good. It is well written and clearly structured, and the conclusions are substantiated by the data (with one notable exception, see below). However, the authors need to address a few issues before the manuscript can be recommended for publication.

Major issues:

– Line 229: ASR evaluation (and following)

One major issue of this study is the usage of a poorly defined "unsintered LSC" film as a current collector.

As the authors state, LSC is a highly active air electrode material. Considering the very thin PLD films, it is not unreasonable to assume that also the LSC "current collector" could contribute to the electrochemical reaction. This can be assumed if the active length of the electrode nanostructure is larger than the layer thickness. The authors do not attempt to calculate the active length, nor do they show thickness-dependent ASR measurements to demonstrate that the nano-LSC layer is thicker than the active length.

Furthermore, the fact that the "unsintered" LSC can be used as a current collector means that the electrical resistance is reasonably low. From this, it can be concluded that sufficient interparticle connectivity exists in this "unsintered" paste and that electrochemical activity can be expected. Applying this LSC paste on the dense LSCF-GDC composite already leads to very good ASR values, which is not intuitive if one were to examine a dense functional layer (which has a fairly low cathode / gas surface area compared to a porous layer) and an inactive current collector.

As a case in point, a recent work by Udomsilp et al. showed very similar SOFC performance to what is shown in this manuscript, using an "in situ activated" (i.e., "unsintered") LSC cathode in a metal-supported fuel cell.[1] In contrast, Hayd et al demonstrated that the LSC paste they used had a much higher ASR than their active LSC layer.[2] The key difference between both pastes is the particle size, with Hayd using very coarse particles and Udomsilp using very fine particles. Therefore the authors need to supply more information about the LSC paste and layer, since they give essentially none in

the manuscript.

It is also not clear why the authors chose a completely unrelated cathode (sintered LSCF on dense LSCF) as the baseline/reference. The logical choice for the baseline in this study would be the "unsintered" LSC paste directly on the GDC electrolyte, without an interlayer. This would allow a direct evaluation of the effect of the interlayers, although the paste can be expected to have a better contact with the porous LSC layer (as it can infiltrate this structure a little when the binder becomes liquid before burn-out) than with the GDC electrolyte.

For publication, it is recommended to provide a baseline measurement with "unsintered" LSC as the only electrode to judge the effect of this layer, and to provide the thickness dependence of the ASR values of the nano-LSC layer to investigate whether the "unsintered" LSC layer can be expected to play a role for the electrochemical activity. This is also important for the evaluation of the cell tests, and help clarify the effect of the current collector on cell performance.

– Line 278: Anode-supported cell fabrication and characterization

The authors need to restructure this section into an evaluation of cell performance that reflects physically meaningful values, rather than the fairly arbitrary "maximum power density" (MPD).

The MPD is a meaningless value for SOFCs. SOFCs are valued for their high efficiency (which the authors also mention in their introduction), and not for their high power density. For an SOFC, a high power density means a low cell voltage, which is tantamount to a low cell efficiency. Klotz et al. have provided useful guidelines for the characterization of SOFCs, recommending to specify the voltage at 1 A/cm², ideally in 60% H₂O / 40% H₂ to minimize the sensitivity of the measurement to humidity variations and self-heating of the cell.[3] To cite Klotz et al. directly from above reference: "The maximum power density (maximum product of voltage and current density in the i/v curve) is irrelevant to SOFCs. It is not a stable operating point (usually < 0.6 V) and the efficiency of the cell in this operating point is very poor."

We therefore recommend to evaluate cell performance using meaningful indicators, such as the cell voltage at a realistic current density (like 1 A/cm²), or the current value at 0.8 V or even 0.9 V. We furthermore note that specifying MPDs for cells with very small active area renders comparisons to larger cells impossible, since even a normal single cell with an active area of 16 cm² usually cannot be tested beyond 2 A/cm² due to hardware limitations. The MPD of such a cell would be limited by the test bench and not the cell itself. This is the case in this study as well: the authors choose a smaller electrode area precisely because their test bench limits the MPD. In this context, the MPD is not a performance indicator of the cell, but rather a characteristic of the test setup, and therefore contains essentially no information about the cell (other than that it shows high performance).

– The authors write: (lines 371-376)

"The change from LSC paste to Pt paste as current collector enabled the attainment of a significantly higher maximum power density at 700 °C, although using LSC paste as current collector is shown to exhibit better performance compared to Pt at lower temperatures (see Supporting Information, Fig. S5, S6, S7). This suggests that the choice of material to be used for current collection is an important consideration for the evaluation of performance of thin film cathodes^{20,21}."

The higher performance of Pt at 700 °C and lower performance below that suggests that the role of this layer is far more complex than simple current collection. As mentioned above, at least the LSC paste can be expected to play an active role in the electrochemical reaction due to the very thin PLD layer. The Pt paste can be expected to infiltrate the LSC porous structure. Higher performance at 700 °C and lower performance below 700 °C suggests a change in activation energy, which is an indicator that the physical processes taking place in the cell are influenced directly by using Pt. Furthermore, Pt

paste may be interesting to achieve high power densities in the laboratory, but has no application potential. The authors have to check and detail this point.

– On the active area:

This is the major shortcoming of this paper. The fact that the area-specific (!) ohmic resistance of the cell decreases significantly (almost by a factor of 4 at 700 °C) when reducing the active area from 0.785 cm² to 0.283 cm² mainly shows that these measurements are not reliable. If the measurement was reliable, the ASR should be the same for both cells.

“Reducing the active electrode area, on the other hand, enabled further reductions in the ohmic resistance value which approximates that calculated using the reported ionic conductivity of YSZ ($\sim 10^{-2}$ S cm⁻¹)³⁸ and its given thickness. We speculate that the lowering of the ohmic resistance may be attributed to the number of structural defects existing in the relatively thin layers of YSZ and GDC, which would be conceivably lower over a smaller electrode area.”

Have the authors considered the implications of this statement, if this were the reason, for larger cell areas? If the ohmic ASR scales with cell area, technically relevant cell areas (~ 80 cm²) would yield very low performances, as the ohmic ASR would increase over 100 times (if the scaling in this manuscript held true). The authors are encouraged to search for the reason for this discrepancy, as it very likely that it is related to artifacts due to the very small cell area.

“With the goal of extracting the highest possible maximum power density that can be achieved using our novel cell architecture,…”

Here, the authors admit that the entire point of this part of the manuscript (lines 367 - 435) serve no other purpose than to obtain a large number that can be used to “make headlines”. As mentioned before, MPD have no practical relevance for SOFCs and are strongly influenced by the active sample area and test bench limitations. The MPD given in this work is also not a physically relevant value, since it is limited by the measurement setup – the “maximum” of the power density curve in Figure 7 has not been reached. If the active area was reduced again, likely the MPD would increase further.

The main problem of this part, however, is the fact that the excellent performance of sample 8 at very high current densities is likely the result of self-heating due to the high current density concentrated in a small area. SOFC current-voltage curves typically have 3 regimes

- i) a non-linear regime with negative curvature at low current, where the non-linearity is due to the activation overpotential (which diminishes with increasing current);
- ii) a linear regime dominated equally by electrode kinetics and ohmic resistance;
- iii) and another non-linear regime (with positive curvature) due to kinetic limitations of gas diffusion (gas diffusion polarization).

The authors confuse regimes I and III in line 351, when stating that positive curvature of the IV-curve indicates activation polarization, when in fact it indicates diffusion polarization.

Regime iii) is mostly avoided in cell tests by using very high reactant fluxes and low fuel utilization. However, there is no electrochemical explanation for the behavior shown in Figure 8, where the slope of the IV-curve becomes continuously smaller with increasing current and a linear regime is not reached even at 8 A/cm². The most likely explanation here is self-heating of the sample due to the high current density.

Consider the differences between samples 7 and 8. As written in the manuscript, the cell architecture and contacting are the same, and the only difference is the active area and the gas flow, both of which are smaller in sample 8. Comparing Figure S6 and Figure S8 (which show sample 7 and 8 respectively, contacted with Pt in both cases), it is immediately obvious that the IV-curves in Figure S6 all show the behavior described above, i.e. a non-linear regime at low current followed by a linear regime. Even at a current density of 3 A/cm², the IV-curves are linear at all temperatures.

Compare Figure S8, where all IV-curves are non-linear for all temperatures and all current densities.

This cannot be an effect of the sample itself, because the sample has the same setup and so the activation overpotential should be visible in the same current range.

A further indication can be seen from the impedance spectra. Figure S8 shows the EIS at OCV and at 0.75 V for sample 8. A detailed examination reveals that the ohmic resistance (the high frequency intercept) depends on the cell voltage, which is unphysical in itself (since the definition of "ohmic" resistance is that it is independent of current or voltage). For instance, the ohmic ASR at OCV and 700 °C is the same as that at 650 °C and 0.75 V. This implies that the ionic conductivity in the electrolyte is the same at both conditions, which implies that at 0.75V ($\sim 2 \text{ A/cm}^2$) the local cell temperature is already 50 °C higher than measured by the thermo-couple. Guk et al have shown that thermocouples close to the sample are insufficient to detect the true sample temperature, and localized heating effects will be even worse in this regard.[4]

It would be best to completely remove this part of the manuscript. The entire point of this paragraph (maximizing MPD) has no physical or technological meaning, since the power output of the cell is likely the same, but only the measurement restrictions have been modified. Furthermore, experimental artifacts cannot be excluded from these measurements, especially significant self-heating due to the localized current, and what is truly measured are simply the experimental constraints (current limitation of the test bench).

Minor issues:

Lines 33-34:

"... of unprecedented values of maximum power densities..."

The maximum power density is not a suitable descriptor of SOFC performance, as described in more detail above. However, very similar SOFC performance has recently been demonstrated by Udomsilp et al using a metal supported cell.[1] The term "unprecedented" is therefore not suitable.

Lines 44-46

"One of the main drawbacks of SOFCs is the relatively high operation temperatures required, which results to several issues such as restriction of allowable materials, thermally activated degradation and instability over long term operation7–10."

The authors are very unspecific here, and this is not sufficient. There exists a commercially available SOFC which is operated at 600 °C (Ceres power, UK). This is not even "relatively" high when compared to the hot zones in combustion engines. Anode-supported SOFCs are routinely operated at 700 °C, and even electrolyte-supported cells are operated at 850°C, permitting the use to stainless steel interconnect materials. The materials selection is not really a challenge any longer for SOFCs. Regarding long-term operation, anode-supported cells from Forschungszentrum Jülich have demonstrated long-term operation for 93,000 hours.[5] Multiple companies have demonstrated 10's of thousands of hours under real operation conditions. Degradation rates below 0.5% / 1000 h (voltage degradation in galvanostatic operation) are routinely demonstrated in systems these days. The introduction to an SOFC paper in a high-profile journal should reflect the reality of SOFC development worldwide today, and not repeat academic talking points. The main issue for SOFCs today is the high system cost. This can be alleviated by cheaper production routes, and by increasing the power density of the cells (resulting in smaller systems for a given power output). Obviously, material selection, degradation and lifetime will always remain an issue, but they are no longer hurdles to commercialization.

Lines 46-48

"To address these issues, several studies focused on lowering the operating temperature by developing novel cathode materials which exhibit low area specific resistance (ASR) and high oxygen exchange properties even at relatively low temperatures11–14."

Again, this is too general. While thermally activated degradation mechanisms would be alleviated by a lower operation temperature, other issues may arise. For instance, internal reforming of carbo-hydrate fuels (a major advantage of SOFCs over other fuel cells) needs a high operation temperature, and may suffer from lower operation temperature. Likewise, stainless steel interconnects suitable for 700 °C can show corrosion at intermediate temperatures.

It would be better if the authors focused on improving the performance, which directly and beneficially impacts system cost, as a driver for cell development.

Lines 56-59:

"Nanoengineered cathodes prepared by advanced thin film techniques such as pulsed laser deposition (PLD) have achieved extremely low ASR values as well as high oxygen exchange properties superior to those of conventional cathodes prepared by wet deposition techniques^{13,15,16}."

This statement is not accurate. The best ASR value in Figure 4 is from a study by Hayd et al, who used a wet deposition technique (a sol-gel technique) for the preparation of a thin LSC electrode. Such techniques are easily scalable and much cheaper than vacuum processes such as PLD.

I suggest to sharpen the point of this sentence by using particle- or suspension-based production techniques like screen-printing as a comparison, since I suspect that the statement about nanoengineering is correct in that comparison. But there are other nanoengineering approaches (e.g. sol-gel techniques, electrostatic spray deposition) that are not suspension-based, but can be called "wet" since they are solution-based.

Lines 63-66_

"This has the benefit of enhancing the cathode/gas and cathode/electrolyte interfacial area density and facilitating the oxygen adsorption and oxide ion transfer, respectively, leading to significant decrease of the polarization resistance values."

It is wrong to say that perovskite/ceria composites inherently increase the cathode/gas interfacial density. This is a matter of the microstructure, not the material selection. As a matter of fact, a composite of an (mostly ORR inactive) ionic conductor and an ORR active perovskite will inherently decrease (!) the cathode/gas interfacial area, compared to a pure perovskite, when the microstructure is the same. This is trivial.

They do inherently increase the cathode/electrolyte interfacial area (again, for a given microstructure).

And so the question of whether a composite is beneficial or not becomes the question of what the rate-limiting factor is for the given cathode chemistry and microstructure. If the oxygen exchange reaction is limited by the cathode surface area, a composite will decrease performance. If it is limited by charge-transfer to the electrolyte, performance will increase.

However, doped ceria powders typically have lower sintering activities compared to LSCF or LSC. Therefore, adding ceria changes the manufacturing of composites with regard to pure perovskites, even when the nominal production parameters (e.g. sintering temperature) remains the same. The problem in this field is related to the nature of academic publishing: when performance can be increased through fabrication of a composite, this is publishable. When performance is decreased by the composite, it is not publishable and therefore not well known.

Lines 70-72:

"Another critical issue is how to tailor cathode nanostructures to be gas permeable and somehow retain this type of structure even at high operating temperatures."

It is interesting that the authors consider this a problem. Typically, gas diffusion in cathode layers is not an issue when using air as oxidant, and only becomes noticeable in electrode performance at lower pO₂ values. Furthermore, the nanostructures are very thin, so that gas diffusion polarization is likely small even for low porosity. Do the authors have some references where this has been shown to be a problem?

The concern that the structures may not remain gas-permeable at higher temperatures is also

counterintuitive. As the authors show in this paper, nanostructures tend to coarsen at elevated temperatures, which actually increases porosity. The likelihood of the nanostructure sintering into a dense layer is very small, given that this is very difficult to achieve under the constraints of the rigid support.

I suspect that the authors did not mean to refer only to gas diffusion when discussing the coarsening of the nanostructures. In that case, this part should be written with improved precision.

Lines 74-77:

"In addition, due to the intrinsically poor lateral conductivity of thin film cathodes, the cell performance of these materials has not yet been fully optimized in terms of appropriately selecting current collectors which would enable good electrical contact with such nanometer-sized grains^{20,21}."

It would be good to point out here that the poor lateral conductivity is primarily a result of the columnar microstructure obtained in the gas-phase deposition process, and not the thickness. In addition, this sentence is to be understood as pre-shading the result that Pt-paste provides the best contact. It bears pointing out that whatever the best solution for contacting SOFCs is, Pt paste is not it due to the extremely high price.

Line 204:

What is a semi-open pore?

Lines 206-209:

"..., since the as-grown film microstructure has been initially tuned to contain porosities, this strategy seems effective in preventing the excessive grain sintering and densification which would otherwise occur for densely packed nanoscaled structures²⁶"

This paragraph is somewhat inaccurate. The primary reason why the nanostructure retains its porosity seems to be that the structure grows in distinct pillars that are isolated from each other. It is because of the lack of contact area between pillars that no densification takes place, since densification requires mass transport via diffusion, and therefore direct contact. Each individual pillar clearly shows densification at 700 °C, as shown in Figure 3. However, as the pillars densify, the distances between the pillars increase.

The increasing porosity at elevated temperature is therefore not a consequence of the presence of porosity alone, but of the particular shape of these pores..

Lines 288-289

"Due to the high sinterability of the nanocomposite particles obtained, the fine AFL structure could be maintained even after high-temperature co-sintering."

This is contradictory. A powder with a high sinterability would not be expected to maintain porosity after high-temperature sintering when allowed to shrink freely (as it is the case here).

Line 346

"High OCV values of >1.1 V were obtained at all temperatures, attributed to the excellent integrity of the thin YSZ electrolyte and dense GDC interlayer employed in the cell architecture."

It would be nice to simply provide the expected value of the OCV for these measurement conditions. Furthermore, the gas tightness of the setup (cell and gasket) should not be judged when using humidified fuel, but dry fuel.

Lines 347-350:

"The overall trend shows that employing the LSCF-GDC nanocomposite layer alone (Sample 5) is insufficient to enhance the maximum power density and moreover, it shows the lowest values among the samples at all temperatures evaluated."

The MPD is not enhanced compared to which reference? The authors do not give any reference here.

Lines 350-535

"Unlike the other two samples, the I-V dependence for this sample also shows a positive curvature, indicating that electrode activation polarization is important³⁷. This may be ascribed to the relatively dense microstructure of this layer where gas diffusion is expected to be limited."

Here, the authors confuse activation polarization with gas diffusion polarization. The two phenomena are distinct and also result in opposite curvatures of the IV-curve.

[1] D. Udomsilp et al., "Metal-Supported Solid Oxide Fuel Cells with Exceptionally High Power Density for Range Extender Systems," *Cell Reports Physical Science*, vol. 1, no. 6, p. 100072, 2020/06/24/ 2020, doi: <https://doi.org/10.1016/j.xcrp.2020.100072>.

[2] J. Hayd, L. Dieterle, U. Guntow, D. Gerthsen, and E. Ivers-Tiffée, "Nanoscaled La_{0.6}Sr_{0.4}CoO_{3-δ} as intermediate temperature solid oxide fuel cell cathode: Microstructure and electrochemical performance," *Journal of Power Sources*, vol. 196, no. 17, pp. 7263-7270, 2011, doi: [10.1016/j.jpowsour.2010.11.147](https://doi.org/10.1016/j.jpowsour.2010.11.147).

[3] D. Klotz, A. Weber, and E. Ivers-Tiffée, "Practical Guidelines for Reliable Electrochemical Characterization of Solid Oxide Fuel Cells," *Electrochimica Acta*, vol. 227, pp. 110-126, 2017/02/10/ 2017, doi: <https://doi.org/10.1016/j.electacta.2016.12.148>.

[4] E. Guk, J.-S. Kim, M. Ranaweera, V. Venkatesan, and L. Jackson, "In-situ monitoring of temperature distribution in operating solid oxide fuel cell cathode using proprietary sensory techniques versus commercial thermocouples," *Applied Energy*, vol. 230, pp. 551-562, 2018/11/15/ 2018, doi: <https://doi.org/10.1016/j.apenergy.2018.08.120>.

[5] Q. Fang, L. Blum, and D. Stolten, "Electrochemical Performance and Degradation Analysis of an SOFC Short Stack for Operation of More than 100,000 Hours," *ECS Transactions*, vol. 91, no. 1, pp. 687-696, July 10, 2019 2019, doi: [10.1149/09101.0687ecst](https://doi.org/10.1149/09101.0687ecst).

Dear Reviewers,

We would like to thank you for your critical assessment of our submission. We have carefully considered all comments received and have modified our manuscript correspondingly. Additional data and revised plots are provided to clarify and resolve the issues pointed out in the original manuscript, as well as overall restructuring of the discussion. With these revisions, we hope that we have satisfactorily addressed all concerns, and that the current version is now acceptable for publication in Nature Communications journal.

Below please find our point-by-point response to the comments received. Our responses are highlighted in red.

On behalf of the co-authors,

Katherine Develos-Bagarinao

AIST, Japan

REVIEWER COMMENTS

Reviewer #1 (Remarks to the Author):

In this manuscript, Katherine Develos-Bagarinao et al. from AIST, Japan reported the achievement of ultrahigh power density at reducing temperature (550-700oC) by using a thin-film YSZ based electrolyte (~ 2um) with (~2um) GDC buffer layer and nano engineered cathode layer with one thin LSCF+GDC self-assembled layer (~0.3um) with ordered structure and a porous nonporous LSC (~1um) of cathode, as prepared by PLD method, which was claimed to demonstrate a peak power density of 3.7 and 4.7 W cm⁻² at 650 and 700oC, respectively, which should be the record-breaking performance in the world. I believe the main creativity of this research is the the highly ordered self-assembled nano composite LSCF+GDC interlayer, and the most attractiveness of this research is the the ultrahigh performance, which suggests YSZ could also be used in IT-SOFC with highly attractive low-temperature performance (>1.6W cm⁻² at 600oC). If the results are reproducible, it may change our thinking of development of IT-SOFC that doped ceria electrolytes and protonic electrolytes that are believed to show higher ionic conductivity at lower temperature are more suitable for IT-SOFCs. However, several critical issues should be explained or clarified before I can suggest the potential publication of this manuscript in Nature Commun.

- 1) As we know, PLD was intensively used to prepare thin film electrolyte or electrode in SOFCs more than 5 years ago, so the concept of use of PLD in SOFC is not new. The main creativity is the preparation of a highly ordered LSCF+GDC dense interlayer. However, very recently, Yufei Song et al. reported the self-assembly concept in the preparation of composite electrode for SOFC (YUfei Song et al. Self-Assembled Triple-Conducting Nanocomposite as a Superior Protonic Ceramic Fuel Cell Cathode, Joule 3

(11), 2842-2853) what is the difference between the two self-assembled concepts should be explained. Otherwise, the novelty of this research will be greatly reduced.

First of all, we would like to thank you for your positive feedback on our submitted work.

In the manuscript we have already cited several references where PLD was specifically used to prepare thin films and nanocomposites for SOFC applications. We agree that it is common knowledge that the concept of using PLD for SOFC materials is not new, but it is also fairly known that most of the work done in this area has been limited to model structures which are not necessarily applicable nor realizable in practical SOFC cells, and even so, the power output of these cells are still relatively low. Our work aims to bridge the gap by providing results where the actual thin films are used in practical anode-supported cells, and also by showing that the achievable power density values surpass those of conventional materials.

Regarding the paper by Song et al.: Firstly, it is not clear why the reviewer would refer to this paper about PCFC, because although they are also based on solid oxide materials, they are based on proton conducting mechanism and operated at relatively lower temperatures, unlike SOFCs which are mainly based on oxide ion transport and operated at relatively higher temperatures. It should also be pointed out that in the paper, the nanocomposite was prepared using a “one-pot sol-gel” method, where the resulting crystallite sizes are in the range of ~100 nm. Due to the nature of the fabrication method adopted by Song et al., the “self-assembly” microstructure described in their paper could not achieve the same degree of order and dimensions as nanocomposites with high aspect ratio (e.g. vertically aligned nanocomposites or “VAN” structures) which are easily achieved using PLD. The pulsed nature of the deposition process, i.e., providing a very high concentration of evaporated species with high energy within a typically short duration of time (~ns), allows for the preferred growth along the normal direction even for a complex binary phase composition. In our study, the LSCF-GDC nanocomposite was prepared using PLD, the crystallite sizes are in the range of ~2-5 nm (at least two orders of magnitude smaller compared to the nanocomposites by Song et al.), in addition to exhibiting a very high aspect ratio and ordering in the nanoscale. This kind of self-assembly would be impossible to achieve using wet deposition techniques.

- 2) Typically, the effective layer thickness of a cathode in SOFC is around 10 μm , this is the reason that most SOFC cathode is prepared to be more than 10 μm in thickness. However, in this study, the authors just prepared the nonporous LSC layer of 1 μm , even including the LSCF+GDC layer, it is also only 1.3 μm , far less than 10 μm . It is recommended that the authors should increase the cathode thickness to larger to see if any beneficial effect, although the cathodic performance base don't he results are already very good

We investigated the thickness dependence of the nanoporous LSC layer (see the response to Reviewer 3) but did not see any significant thickness dependence in terms of performance. On the other hand, it is debatable that using PLD to deposit thicker layers would be of any practical merit, due to the relatively low film rate it would take

hours to prepare films of several microns in thickness.

- 3) One of my major concern is the accuracy of the measurement. For thin film electrolyte, the gas leaking between the anode and cathode chamber is very easily to appear, while the direct non electrochemical oxidation of the mixed H₂ and O₂ will make the cell temperature higher than the furnace temperature, that will make the cell power output higher than the actual value at set temperatures. The authors used 2μm YSZ electrolyte and 2μm GDC buffer layer, both of them will contribute to the ohmic resistance of the cell. The maximum power density of a cell can be calculated based on $PPD = OCV^2 / 4 * (\text{total cell resistance})$. Since the cell OCV is around 1.1 V to achieve a peak power density of 4.7 and 3.7 W cm⁻², it means the cell total resistance should be 0.0644ohm cm² and 0.0818 ohm cm², respectively at 700 and 650 oC. However, based on EIS in Fig.7, the value at 700oC matched well, but the value at 650oC (0.125 ohm cm²) is much larger. At lower temperature such discrepancy is even more obvious. The authors should provide a reasonable explanation.

The correct value for the maximum power density at 650 °C should have been 3.4 W/cm². We apologize for the mistake in quoting the correct value.

In the revised version, in response to the suggestion from Reviewer 3, we have removed references to the MPD and focused on relevant values based on the current densities obtained at an operating voltage of 0.7 V.

As for the discrepancies pointed out by the reviewer regarding the total resistance R_t, these are easily resolved when we consider that the characteristic I-V is non-linear, which means that the predicted R_t values calculated using the maximum power density and OCV would be different from the actual R_t measured. On the other hand, if we derive the I-V slope in the vicinity of 0.75 V, the value that corresponds to the R_t matches well with that derived from the EIS data, as follows:

○650□

R_t, ASR (from EIS) @0.75V : 122 mΩcm²
Slope of I-V@0.75V : 113 mΩcm²

○600□

R_t, ASR (from EIS) @0.75V : 255 mΩcm²
Slope of I-V@0.75 : 241 mΩcm²

○550□

R_t, ASR (from EIS) @0.75V : 530 mΩcm²
Slope of I-V@0.75 : 508 mΩcm²

The R_t values shown in the figure below reflect the values derived from the EIS measurements.

- 4) As to the stability/durability issue, although the authors tested for a period of 2500C at 7000C at 1.0A cm-2, the cell power output is fairly stable. However, if we compared the results in Fig. 7 and Fig. 8, at the initial test, no obvious concentration polarization was observed at high current density, however, at the end of the stability test, based on the I-V curve, large drop in cell voltage at high current density suggests the appearance of concentration polarization. It implies, the electrode is likely sintered. SEM observation of the tested cell after the durability test should be provided.

The reviewer is correct in pointing out that difference in the I-V characteristics after the stability test. SEM observations were conducted for the tested cell, and indeed we have confirmed sintering of the electrodes. However, by including these results we feel that it would deviate from the main focus of this publication (i.e., performance), and we prefer to include the microstructural evaluation in a follow-up publication which will be focused on the issue of long-term stability.

- 5) The advances in cathode development is very fast during the past several years, however, the authors' reference is mostly before 2018. I would suggest the update of references by providing more recent publications .

Additional references have been added.

If above comments are well addressed, I would suggest the publication of this manuscript in Nature Commun.

Reviewer #2 (Remarks to the Author):

This manuscript with title "Nanoengineering of cathode layers for solid oxide fuel cells to

achieve superior power densities" presents a very significant break-through in IT-SOFC power density by using nanoengineered cathode electrode layers, namely LSCF-GDC nanocomposite transition layer and LSC nanoporous layer. The data and image quality fulfills the requirements of Nature Communication. Despite the significance of the manuscript, I would request for some clarifications and major revisions especially for justifying the high performance reported by the authors:

We would like to thank the reviewer for their positive assessment of our manuscript.

1. Page 5 Line 118-121: Can the authors explain how 100 times or more cathode/electrolyte interfacial area is calculated? Which baseline are you comparing to?

This is in comparison to a baseline sample where the interfacial area only exists between a pure LSCF layer and a GDC electrolyte. The missing description has been added to the text in p.5.

To explain the estimated increase in interfacial area, take for example a unit area of 100 nm x 100 nm as illustrated below. For a pure LSCF layer in contact with a GDC electrolyte we can compute the interfacial area to be 10,000 nm². Consider a nanocomposite of LSCF-GDC where the individual phases exist in nanocolumns of ~5 nm in width and that the total thickness is ~300 nm. Introduction of the GDC phase in the nanocomposite layer will yield 50% interfacial contact with the GDC electrolyte compared to a pure LSCF layer, i.e., the interfacial area at the boundary of the nanocomposite-GDC electrolyte will be reduced to 5,000 nm². However, within the nanocomposite region itself (in-plane), there are now additional interfacial areas in contact between the LSCF and GDC phases, and therefore we should include these in the calculation of the total interfacial area.

Interfacial area: 10,000 nm²

Interfacial area: 5 nm x 5 nm x 10 x 20 = 5,000 nm²

To illustrate this, refer to the next figure showing the top view of a simplified LSCF-GDC matrix where the phases are evenly distributed in a checkered pattern over a 100 nm x 100 nm area. Each phase is assigned to a square representing a 5 nm x 5 nm x 300 nm column.

Here, the GDC phase is represented by the light blue squares whereas the LSCF phase is represented by the gray-, black-, and brown-colored squares. The inner squares representing LSCF sharing a boundary on all sides with the GDC phase are shown in gray, the outermost LSCF which share three boundaries with the GDC phase are represented in black, and the corners which share only two boundaries with GDC are shown in brown. For a given thickness of ~ 300 nm, the interfacial area of an individual side will be $300 \text{ nm} \times 5 \text{ nm} = 1500 \text{ nm}^2$. Considering the number of shared boundaries, we can obtain the total interfacial area as follows:

Inner	$1500 \text{ nm}^2 \times 9 \times 18 \times 4 = 972,000$	nm^2
Edges	$1500 \text{ nm}^2 \times 9 \times 4 \times 3 = 162,000$	nm^2
Corners	$1500 \text{ nm}^2 \times 2 \times 2 = 6,000$	nm^2
Interface		$5,000 \text{ nm}^2$
Total		$1,145,000 \text{ nm}^2$

The above results show approximately 100 times higher total interfacial area compared to that of a pure LSCF layer on GDC electrolyte (only $10,000 \text{ nm}^2$). As can be seen, the most significant contribution to the total interfacial area comes from the existence of a high density of interfaces between LSCF and GDC within the nanocomposite layer. From this analysis, it would be easy to see that increasing its film thickness would result to even further increases in the total interfacial area.

- Page 8 Line 191: Authors mention that "it is essential that the cathode structure is sufficiently porous to fulfill the gas permeability requirement." Similar to nanoporous LSC layer, to my understanding LSCF-GDC is also an electrode. Can the authors explain the reason why the LSCF-GDC nanocomposite layer does not require porosity for gas permeability? Wouldn't it be more beneficial if the LSCF-GDC nanocomposite layer is also porous so that electrochemical reaction occurs even closer to the GDC electrolyte?

The LSCF-GDC nanocomposite is part of the cathode structure but essentially acts as a transition layer and plays an important role of improving charge transfer across the electrode-electrolyte through its high-density interface. The LSCF-GDC nanocomposite

was prepared dense for two primary reasons: 1) to achieve the highly ordered self-assembled nanostructures, and 2) to clearly differentiate its function from that of the nanoporous LSC layer. By tuning the deposition parameters, it is of course possible to make nanocomposite layer porous as well; we agree with the reviewer that introducing porosity will be beneficial somewhat even though the additional porosities will likely decrease the amount of LSCF-GDC interfaces compared to a fully dense structure. In fact, this is part of our ongoing work on optimizing the nanoengineered cathodes by modifying the microstructure. These results will be reported in a future publication.

3. Page 10 Line 246-248: This is not a fair comparison because Sample 4b does not have GDC. Composite electrode such as LSCF-GDC is known to improve cell performance compared to single phase electrode such as LSCF even with conventional wet deposited layer, due to the improved ionic conductivity. The significance of LSCF-GDC nanocomposite layer fabricated by PLD in comparison to a conventional LSCF-GDC composite layer engineered by conventional methods is not evident here.

The data representing LSCF-GDC composite layers prepared by conventional methods are already included in Fig. 4 (the corresponding data are shown in gray-color open symbols). As prepared by conventional methods, the composites only showed marginal improvements over their respective LSCF counterparts (also shown in Fig. 4, gray filled symbols), suggesting that the expected improvement is not consistently observed for these types of samples. On the other hand, we drew the comparison with Sample 4b (Sample 5 in the revised manuscript) to primarily show that using a LSCF-GDC nanocomposite at the interface with the GDC electrolyte leads to further reduction in ASR as compared to a pure LSCF film prepared using the same deposition conditions.

4. Page 15 Line 351: Can you please explain in detail here how a positive curvature is correlated with activation polarization? It would be helpful if you can also show dV/dI curve and explain the difference among these three samples.

The positive curvature at high current densities is correlated with the gas diffusion polarization, not activation polarization as initially stated. This has been corrected in the revised version.

5. Page 15 Line 351-352: Following the above question, doesn't this dense microstructure of LSCF-GDC nanocomposite layer also exist in Sample 7? Why does Sample 7 not showing a positive curvature? Please explain.

The positive curvature is also evident for Sample 7 (Sample 8 in the revised version), albeit at relatively higher current values.

6. Page 15 Line 370: This is one of my biggest questions for this manuscript. Is the significant higher performance due to the smaller active area or is it due to the use of Pt paste? Can the authors provide performance of a cell with 0.785 cm² active area but

with Pt paste as current collector, so that we know the practical maximum power density for a single cell with your technology?

The data was already included in the original Supplementary (Fig. S6). As this data was somehow missed, and to make the comparison clearer, we added the data side-by-side to that of a sample having the same active area of 0.785 cm^2 but evaluated using LSC paste in the new Fig. 7 of the revised manuscript.

For the same active area, the use of Pt paste as current collector seems to lead to better performance than for LSC paste at $700 \text{ }^\circ\text{C}$, but they show comparable performances at lower temperatures. As explained in the main text, the use of LSC paste likely dominates the performance of the cells by reacting with the nanoporous LSC at $700 \text{ }^\circ\text{C}$, but this may not be the case for the Pt paste.

Also, what is the cell total area? A very small active area to total area ratio is not realistic in the stack due to waste of extra anode materials. Please justify and explain.

The total cell area is 3.801 cm^2 ($r = 1.1 \text{ cm}$). The relatively smaller active area of 0.785 cm^2 is actually limited by our measurement rig, but for our purposes it is sufficient to demonstrate the performance of the cell at the laboratory scale.

Realistically, it is possible expand the active area further by preparing the thin film cathodes over a wider area of the cell. Although our proposed strategy has potential use for practical cells, admittedly there are still issues with upscaling which should be addressed and consequently developed. We do not claim that keeping the active area small is an economical route for practical cells, only that for this study it is limited by the capability of our test bench.

7. Page 16 Line 378-381, "We speculate...smaller electrode area". This argument doesn't make sense. The number of structural defects in YSZ and GDC in a unit area should be equal as you are using PLD for fabrication. When increasing the active area of the electrode, the area specific resistance should therefore be equal, theoretically. Again, a sample with the same configuration of Sample 8, but with Pt paste is necessary to separate the contribution of Pt paste and the contribution of smaller active area. Please provide additional data to justify.

The apparent discrepancies in the ohmic resistance values have been corrected by taking into account inductance effects in the impedance spectra. In the revised version, we show that the ohmic resistance values are indeed comparable for the samples regardless of the active area. Again, for clarity the samples having the same configuration but with different current collectors are now shown side-by-side in the new Fig. 7.

8. Page 16 Line 384: "measured open-circuit voltages (OCV) at $700 \text{ }^\circ\text{C}$ is 1.093 V which is almost equal to the theoretical OCV". This is not correct. With 3% humidified H_2 and dry air, the theoretical OCV of the cell at $700 \text{ }^\circ\text{C}$ should be equal to 1.120 V , which is much higher than 1.093 V . One could argue that this cell has some leakage which cause

local temperature of the cell higher than 700 C, resulting in a high power density value. Please explain.

The OCV value of ~1.1 V is slightly lower than the theoretical value of ~1.12 V but still lies within the range of experimental values reported in literature, i.e., ~1.05-1.1 V (see e.g. Refs. 1-4 below), thus allowing for a reasonable comparison of data. We also consider that this is a reasonably high value for the OCV, considering that the YSZ electrolyte is only a few microns in thickness.

As mentioned in the Methods section, the temperature of the cell is carefully monitored using a thermocouple located ~1 mm away from the cathode side. As far as we can tell, there are no indications of any significant increases in the cell temperature which may arise from gas leakage.

Nevertheless, similar to the publications referenced below, we concur that it is difficult to totally exclude the possibility of gas leakage (e.g. along peripheries of the anode support which were not sufficiently sealed), which can account for the slight difference of the experimental OCV from the theoretical value, but its association to possible artefacts due to heating is unlikely.

1. F. Han et al., Journal of Power Sources 218 (2012) 157-162, <http://dx.doi.org/10.1016/j.jpowsour.2012.06.087>
2. D. Udomsilp et al 2019 J. Electrochem. Soc. 166 F506, <http://dx.doi.org/10.1149/2.0561908jes>
3. Y.H. Lee et al., Nano Lett. 2020, 20, 2943–2949, <https://dx.doi.org/10.1021/acs.nanolett.9b02344>
4. B.-K. Park and S.A. Barnett, J. Mater. Chem. A, 2020, 8, 11626, <https://dx.doi.org/10.1039/d0ta04280c>

9. Page 16 Line 395, Figure S8: The cell in Fig. S8 seems to have higher power density at 0.75 V, why don't you report this cell in the main text, instead of Sample 8?

Actually, for the cell shown in Fig. S6 (previously Fig. S8), the power density at 0.75 V is 2.74 W/cm², whereas the one in the main text is 2.80 W/cm², which is slightly higher, albeit the difference is quite insignificant in our opinion. No changes have been made.

10. Page 18 Table 2: Once again, we need to first justify the contribution of using a small active area. This table of comparison may not be fair, as we don't know the "active area to cell total area ratio" for these cells in literature. One could argue that the high performance reported in this paper is due to a very small active area. Again, for comparison, a reasonable and realistic active area should be used. By doing that, the readers can then better understand the potential of your technology.

We understand the reviewer's concern about the effect of active area. Unfortunately, our testing facility is not yet able to test larger cells as of this writing. We are planning to prepare cells with larger areas in the future.

Furthermore, in response to the recommendation by Reviewer 3, we updated Table 2 using a more meaningful indicator given by the current density values at 0.7 V, which would be reasonable for comparison across the cells reported in literature.

11. Page 19 Fig. 8b: In Fig. 8b, can the authors also include the performance of the cell before testing, to be compared with data after testing?

The initial performance of this cell is shown by Fig. S6 in the Supplementary Information. This cell shows a performance similar to the cell shown in the main text.

12. Page 23 Line 546-547: Why are the flow rates for fuel gas and air different for Sample 8 from those for Sample 5, 6, 7?

We reduced the flow rates for Sample 9 (smaller area; previously Sample 8) in order to approximate the conditions for Sample 6, 7, and 8 (larger area; previously Sample 5, 6, and 7). However, on a separate measurement we tried decreasing the flow rate to the same levels as those of Sample 9 but confirmed that there was no dependence on the flow rate.

Reviewer #3 (Remarks to the Author):

The manuscript describes an innovative approach to produce functional layers of a solid oxide fuel cell (SOFC) using Pulsed Laser Deposition (PLD). While PLD is (at least in the academic world) routinely used to produce SOFCs, the authors present an interesting approach in fabricating a dual-phase interfacial layer between the electrolyte and the air electrode, and provide evidence that this layer is beneficial for cell performance.

Overall, the quality of the manuscript is very good. It is well written and clearly structured, and the conclusions are substantiated by the data (with one notable exception, see below). However, the authors need to address a few issues before the manuscript can be recommended for publication.

We would like to thank the reviewer for their positive evaluation of our manuscript. More importantly, we appreciate the detailed assessment by the reviewer by not only highlighting the weak areas in the manuscript but also providing helpful suggestions for improvement.

Major issues:

- Line 229: ASR evaluation (and following)

One major issue of this study is the usage of a poorly defined “unsintered LSC” film as a current collector.

As the authors state, LSC is a highly active air electrode material. Considering the very thin PLD films, it is not unreasonable to assume that also the LSC “current collector” could contribute to the electrochemical reaction. This can be assumed if the active length of the electrode nanostructure is larger than the layer thickness. The authors do not attempt to calculate the active length, nor do they show thickness-dependent ASR measurements to demonstrate that the nano-LSC layer is thicker than the active length.

Furthermore, the fact that the “unsintered” LSC can be used as a current collector means that the electrical resistance is reasonably low. From this, it can be concluded that sufficient interparticle connectivity exists in this “unsintered” paste and that electrochemical activity can be expected. Applying this LSC paste on the dense LSCF-GDC composite already leads to very good ASR values, which is not intuitive if one were to examine a dense functional layer (which has a fairly low cathode / gas surface area compared to a porous layer) and an inactive current collector.

As a case in point, a recent work by Udomsilp et al. showed very similar SOFC performance to what is shown in this manuscript, using an “in situ activated” (i.e., “unsintered”) LSC cathode in a metal-supported fuel cell.[1] In contrast, Hayd et al demonstrated that the LSC paste they used had a much higher ASR than their active LSC layer.[2] The key difference between both pastes is the particle size, with Hayd using very coarse particles and Udomsilp using very fine particles. Therefore the authors need to supply more information about the LSC paste and layer, since they give essentially none in the manuscript.

It is also not clear why the authors chose a completely unrelated cathode (sintered LSCF on dense LSCF) as the baseline/reference. The logical choice for the baseline in this study would be the “unsintered” LSC paste directly on the GDC electrolyte, without an interlayer. This would allow a direct evaluation of the effect of the interlayers, although the paste can be expected to have a better contact with the porous LSC layer (as it can infiltrate this structure a little when the binder becomes liquid before burn-out) than with the GDC electrolyte.

For publication, it is recommended to provide a baseline measurement with “unsintered” LSC as the only electrode to judge the effect of this layer, and to provide the thickness dependence of the ASR values of the nano-LSC layer to investigate whether the “unsintered” LSC layer can be expected to play a role for the electrochemical activity. This is also important for the evaluation of the cell tests, and help clarify the effect of the current collector on cell performance.

As recommended by the reviewer, we added a baseline sample of unsintered LSC to Fig. 4. Additional information on the LSC paste are given in the Methods section. A typical SEM image is further shown in Fig. 7(e).

The new data shows that for unsintered LSC paste only (Sample 1), relatively higher ASR values compared to the samples containing any PLD layers below 600 °C were obtained, suggesting that the improvement in ASR performance can be unambiguously attributed to the addition of the thin film cathodes. However, at 600-700 °C, with the exception of Sample 4 (LSCF-GDC nanocomposite + nanoporous LSC), the performance of the LSC paste is comparable to the other samples containing the nanoporous LSC (Sample 2 and Sample 5). From these results, it appears that the performance of the cells having the nanoporous LSC layers becomes comparable to that of the LSC paste. One plausible explanation is that the nanoporous LSC interacts with the LSC paste as the cells are heated, which is quite expected given that this layer was prepared at room temperature and would therefore invariably change

upon heating. On the other hand, the behavior seems to be different for the case of the LSCF-GDC nanocomposite only (Sample 3), which shows relatively higher ASR values compared to Sample 1's in the same temperature range. This may be due to the fact that this layer was heated during PLD deposition and is therefore more microstructurally and thermally stable and less likely to be influenced by the LSC paste compared to the nanoporous LSC. Furthermore, due to its nanosized domains, without the presence of the nanoporous LSC its contact with the LSC current collector is likely not as good.

We examined the thickness dependence of the nanoporous LSC layers by varying its thickness from an as-grown nominal value of $\sim 0.5 \mu\text{m}$ to $\sim 2 \mu\text{m}$ (see representative SEM images below; note that these are only shown for the purpose of review and are not included in the manuscript) but did not find any significant thickness dependence of the ASR performance among the samples. There is also an issue of “grain bunching” with higher thickness of the nanoporous LSC layer, suggesting that it would be inaccurate to compare nanoporous LSC layers simply on the basis of thickness as the films themselves also display different sintering behaviors depending on thickness.

Comparison of the SEM images of the unsintered LSC paste used in this study to that of Udomsilp et al. reveals that their microstructures are highly similar.

To summarize:

1. At sufficiently low temperatures, we can consider that the contribution to the electrochemical activity of the unsintered LSC paste is likely minimal, and the improvement in performance may be directly attributed to the presence of the thin film cathodes.
2. At sufficiently high temperatures, it is difficult to distinguish the effect of the nanoporous LSC layers from the contribution of the unsintered LSC paste, possibly due to their interaction at these temperatures.
3. Compared to the LSC paste baseline as well as other samples, the sample having the combined nanoengineered cathodes (LSCF-GDC nanocomposite + nanoporous LSC) still exhibits the lowest ASR values, indicating that although its overall performance likely includes the contribution of the unsintered LSC paste as well, it nevertheless exhibits the best ASR performance among all samples (also contacted by unsintered LSC paste).

Based on these results, we believe that the use of Pt paste as an alternative current collector is therefore justified.

□ Line 278: Anode-supported cell fabrication and characterization

The authors need to restructure this section into an evaluation of cell performance that reflects physically meaningful values, rather than the fairly arbitrary “maximum power density” (MPD).

The MPD is a meaningless value for SOFCs. SOFCs are valued for their high efficiency (which the authors also mention in their introduction), and not for their high power density. For an SOFC, a high power density means a low cell voltage, which is tantamount to a low cell

efficiency. Klotz et al. have provided useful guidelines for the characterization of SOFCs, recommending to specify the voltage at 1 A/cm², ideally in 60% H₂O / 40% H₂ to minimize the sensitivity of the measurement to humidity variations and self-heating of the cell.[3] To cite Klotz et al. directly from above reference: “The maximum power density (maximum product of voltage and current density in the i/v curve) is irrelevant to SOFCs. It is not a stable operating point (usually < 0.6 V) and the efficiency of the cell in this operating point is very poor.”

We therefore recommend to evaluate cell performance using meaningful indicators, such as the cell voltage at a realistic current density (like 1 A/cm²), or the current value at 0.8 V or even 0.9 V. We furthermore note that specifying MPDs for cells with very small active area renders comparisons to larger cells impossible, since even a normal single cell with an active area of 16 cm² usually cannot be tested beyond 2 A/cm² due to hardware limitations. The MPD of such a cell would be limited by the test bench and not the cell itself. This is the case in this study as well: the authors choose a smaller electrode area precisely because their test bench limits the MPD. In this context, the MPD is not a performance indicator of the cell, but rather a characteristic of the test setup, and therefore contains essentially no information about the cell (other than that it shows high performance).

This point is well-taken, and we therefore revised the manuscript by using a more meaningful indicator given by the current density value at 0.7 V. Table 2 has been updated with the corresponding values.

□ The authors write: (lines 371-376)

“The change from LSC paste to Pt paste as current collector enabled the attainment of a significantly higher maximum power density at 700 °C, although using LSC paste as current collector is shown to exhibit better performance compared to Pt at lower temperatures (see Supporting Information, Fig. S5, S6, S7). This suggests that the choice of material to be used for current collection is an important consideration for the evaluation of performance of thin film cathodes^{20,21}.”

The higher performance of Pt at 700 °C and lower performance below that suggests that the role of this layer is far more complex than simple current collection. As mentioned above, at least the LSC paste can be expected to play an active role in the electrochemical reaction due to the very thin PLD layer. The Pt paste can be expected to infiltrate the LSC porous structure. Higher performance at 700 °C and lower performance below 700 °C suggests a change in activation energy, which is an indicator that the physical processes taking place in the cell are influenced directly by using Pt. Furthermore, Pt paste may be interesting to achieve high power densities in the laboratory, but has no application potential. The authors have to check and detail this point.

The reviewer is not necessarily incorrect in their interpretation in this regard, but given the new results on the investigation on the influence of the LSC paste, we think that the more correct interpretation here is that the LSC paste has very likely limited the performance evaluation at 700 °C due to its possible interaction with the nanoporous LSC.

There is no evidence that the Pt paste infiltrates the nanoporous LSC structure. The high

sinterability of the grains in the Pt paste lead to much larger grains in comparison to those of the thin film layers. The post-test SEM (backscattered electron mode) observations show that the Pt paste can be detached easily from the surface of the films:

Some Pt residues can be observed only on the top of the nanoporous LSC structures but there are certainly no grains in the interior.

□ On the active area:

This is the major shortcoming of this paper. The fact that the area-specific (!) ohmic resistance of the cell decreases significantly (almost by a factor of 4 at 700 °C) when reducing the active area from 0.785 cm² to 0.283 cm² mainly shows that these measurements are not reliable. If the

measurement was reliable, the ASR should be the same for both cells.

The other reviewers also raised similar concerns. We rechecked our impedance data and found that inductance effects have not been considered in the fitting of the spectra. By taking into account the inductance contribution, the calculated area-specific ohmic resistances have been corrected. The results now show comparable ASR values for the cells with different active areas but similarly contacted with Pt paste, as expected.

“Reducing the active electrode area, on the other hand, enabled further reductions in the ohmic resistance value which approximates that calculated using the reported ionic conductivity of YSZ ($\sim 10^{-2}$ S cm⁻¹)³⁸ and its given thickness. We speculate that the lowering of the ohmic resistance may be attributed to the number of structural defects existing in the relatively thin layers of YSZ and GDC, which would be conceivably lower over a smaller electrode area.”

Have the authors considered the implications of this statement, if this were the reason, for larger cell areas? If the ohmic ASR scales with cell area, technically relevant cell areas (~ 80 cm²) would yield very low performances, as the ohmic ASR would increase over 100 times (if the scaling in this manuscript held true). The authors are encouraged to search for the reason for this discrepancy, as it is very likely that it is related to artifacts due to the very small cell area.

We agree that this explanation is inadequate, and we have therefore deleted this part from the manuscript. As explained above, we have already corrected this error and revised the corresponding explanation.

“With the goal of extracting the highest possible maximum power density that can be achieved using our novel cell architecture,…”

Here, the authors admit that the entire point of this part of the manuscript (lines 367 - 435) serve no other purpose than to obtain a large number that can be used to “make headlines”. As mentioned before, MPD have no practical relevance for SOFCs and are strongly influenced by the active sample area and test bench limitations. The MPD given in this work is also not a physically relevant value, since it is limited by the measurement setup – the “maximum” of the power density curve in Figure 7 has not been reached. If the active area was reduced again, likely the MPD would increase further.

We have omitted references to the MPD and included relevant values based on the current densities obtained at an operating voltage of 0.7 V. This is mainly to facilitate the comparison with existing values reported in literature.

Researchers are sometimes guilty of sensationalism in publicizing their findings, but we disagree that this is the case in point here. The only point worth admitting here is that the values derived are among the highest reported but even so, it is highly questionable that merely quoting large numbers would in themselves merit “headlines.” The numbers would only remain numbers if they could not be translated into practical and useful technology, if and when they do, only then would they “make headlines.”

The main problem of this part, however, is the fact that the excellent performance of sample 8 at very high current densities is likely the result of self-heating due to the high current density concentrated in a small area. SOFC current-voltage curves typically have 3 regimes

- i) a non-linear regime with negative curvature at low current, where the non-linearity is due to the activation overpotential (which diminishes with increasing current);
- ii) a linear regime dominated equally by electrode kinetics and ohmic resistance;
- iii) and another non-linear regime (with positive curvature) due to kinetic limitations of gas diffusion (gas diffusion polarization).

The authors confuse regimes I and III in line 351, when stating that positive curvature of the IV-curve indicates activation polarization, when in fact it indicates diffusion polarization.

Regime iii) is mostly avoided in cell tests by using very high reactant fluxes and low fuel utilization. However, there is no electrochemical explanation for the behavior shown in Figure 8, where the slope of the IV-curve becomes continuously smaller with increasing current and a linear regime is not reached even at 8 A/cm². The most likely explanation here is self-heating of the sample due to the high current density.

Consider the differences between samples 7 and 8. As written in the manuscript, the cell architecture and contacting are the same, and the only difference is the active area and the gas flow, both of which are smaller in sample 8. Comparing Figure S6 and Figure S8 (which show sample 7 and 8 respectively, contacted with Pt in both cases), it is immediately obvious that the IV-curves in Figure S6 all show the behavior described above, i.e. a non-linear regime at low current followed by a linear regime. Even at a current density of 3 A/cm², the IV-curves are linear at all temperatures.

Compare Figure S8, where all IV-curves are non-linear for all temperatures and all current densities. This cannot be an effect of the sample itself, because the sample has the same setup and so the activation overpotential should be visible in the same current range.

Thank you for your insightful comments.

If we directly superimpose the I-V-curves shown in Fig. S6 and S8 (Fig. 7b and 8a in the revised manuscript, respectively), the curves appear to overlap with each other, indicating that their performances are comparable at least up to ~3 A/cm².

The described regimes may be typical for the case of porous cathodes, but for the highly ordered, nanostructured cathodes in this study, we think that the behavior may be more complicated. Given its characteristic highly ordered nanostructure, deviations in I-V behavior from that exhibited by porous cathodes is understandable. Is it correct to assume that the nanoengineered cathodes in this study, with dimensions much smaller by at least 2-3 orders of magnitude, in a self-assembled nanostructure, would actually behave as the porous cathodes?

Though we cannot entirely exclude the possibility of self-heating, first of all we can observe that the impedance spectra do not exhibit any significant differences between OCV and 0.75 V, suggesting a possible activation with current density, leading to the observed non-linearity in the I-V curves.

This “anomalous” behavior may be directly correlated to the complexity of the nanostructures developed in this study but also to hitherto unexplored properties which are only realized at the nanoscale. The highly coherent nanostructures in the nanocomposite, for instance, result to significant lattice strains at the interfaces, more so at high operating temperatures. Our ongoing analyses using advanced characterization tools have so far revealed that the electronic structure of the materials comprising the nanoengineered cathodes highly differs from that typically exhibited by bulk samples. Although at this point it is still not entirely clear how these characteristics can influence electrochemical behavior at operating conditions, we contend that just because the behavior does not fit those of conventional cathodes, it does not mean that it should be easily dismissed as an effect of some unknown experimental artefact. Rather, we hope that drawing attention to this “anomalous” behavior of nanostructured cathodes will generate a lot of interest for future studies.

A further indication can be seen from the impedance spectra. Figure S8 shows the EIS at OCV and at 0.75 V for sample 8. A detailed examination reveals that the ohmic resistance (the high frequency intercept) depends on the cell voltage, which is unphysical in itself (since the definition of “ohmic” resistance is that it is independent of current or voltage). For instance, the ohmic ASR at OCV and 700 °C is the same as that at 650 °C and 0.75 V. This implies that the ionic conductivity in the electrolyte is the same at both conditions, which implies that at 0.75V (~ 2 A/cm²) the local cell temperature is already 50 °C higher than measured by the thermocouple. Guk et al have shown that thermocouples close to the sample are insufficient to detect the true sample temperature, and localized heating effects will be even worse in this regard.[4]

It would be best to completely remove this part of the manuscript. The entire point of this paragraph (maximizing MPD) has no physical or technological meaning, since the power output of the cell is likely the same, but only the measurement restrictions have been modified. Furthermore, experimental artifacts cannot be excluded from these measurements, especially significant self-heating due to the localized current, and what is truly measured are simply the experimental constraints (current limitation of the test bench).

On the subject of self-heating: we agree that self-heating could have occurred in the samples, but the more relevant question is, *to what extent?*

Here the reviewer is basing their estimate of the local cell temperature on the supposed coincidence of the ohmic resistance at OCV and 700 °C and that at 650 °C and 0.75 V in Fig. S8. This is a clear misunderstanding, which we suspect is due to the slight misalignment of the impedance plots supplied in the original Supplementary Information (the OCV plot has been slightly shifted to the right, and if the reviewer simply drew a line between the two plots it would look like the points coincided). We reproduce the said plots and overlap them as shown below:

As a case in point, we can clearly see that the ohmic resistance values are actually the same for 650 °C at OCV *and* 0.75 V. The same is true for 700 °C. Therefore, the temperature difference alluded by the reviewer is incorrect.

The possibility of self-heating which might have occurred in Sample 8 certainly deserves further investigation, but if such experimental artefacts were truly impossible to exclude from these types of measurements, then it is just as arguable that any experimental data reported for cells with high power densities in literature inherently includes these effects.

On the other hand, the work by Guk et al. (Appl. Energy 2018) and their more recent publication (Appl. Energy 2020 <https://doi.org/10.1016/j.apenergy.2020.116013>) indeed have shown that there are differences in the measured temperature by thermocouples placed in proximity to the surface compared to those “implanted” into the cathode surfaces. Even so, the differences are only a few degrees and mostly arise due to the inherent spatial distribution across the cell. To quote directly from the 2020 article: “...a principal drawback of thermocouples in SOFC temperature sensing is its inability to measure the electrode temperature with sufficiently high spatial resolution.” This is not necessarily related to any self-heating behavior and is mostly a consequence of the temperature profiles induced by changes in the operating conditions.

Based on these considerations, we made changes into how the results were discussed but opted to retain the results mentioned in this part of the manuscript.

Minor issues:

Lines 33-34:

“... of unprecedented values of maximum power densities...”

The maximum power density is not a suitable descriptor of SOFC performance, as described in more detail above. However, very similar SOFC performance has recently been demonstrated by Udomsilp et al using a metal supported cell.[1] The term “unprecedented” is therefore not suitable.

This has been corrected to:

“...of high current densities at 0.7 V reaching...”

Lines 44-46

“One of the main drawbacks of SOFCs is the relatively high operation temperatures required, which results to several issues such as restriction of allowable materials, thermally activated degradation and instability over long term operation7–10.”

The authors are very unspecific here, and this is not sufficient. There exists a commercially available SOFC which is operated at 600 °C (Ceres power, UK). This is not even “relatively” high when compared to the hot zones in combustion engines. Anode-supported SOFCs are routinely operated at 700 °C, and even electrolyte-supported cells are operated at 850°C, permitting the use to stainless steel interconnect materials. The materials selection is not really a challenge any longer for SOFCs.

Regarding long-term operation, anode-supported cells from Forschungszentrum Jülich have demonstrated long-term operation for 93,000 hours.[5] Multiple companies have demonstrated 10’s of thousands of hours under real operation conditions. Degradation rates below 0.5% / 1000 h (voltage degradation in galvanostatic operation) are routinely demonstrated in systems these days.

The introduction to an SOFC paper in a high-profile journal should reflect the reality of SOFC development worldwide today, and not repeat academic talking points. The main issue for SOFCs today is the high system cost. This can be alleviated by cheaper production routes, and by increasing the power density of the cells (resulting in smaller systems for a given power output). Obviously, material selection, degradation and lifetime will always remain an issue, but they are no longer hurdles to commercialization.

This part has been rewritten as follows:

“Durability and performance stability over long-term operation have been the focus of R&D efforts in both academic and industrial sectors^{4–11}, however, a persistent key issue required for widespread commercialization of SOFC technology is the lowering of the system cost. In addition to improving production routes and efficiency of cell stacks, improving performance by achieving higher power densities has been identified as a strategy to reduce stack size and system cost.”

Lines 46-48

“To address these issues, several studies focused on lowering the operating temperature by developing novel cathode materials which exhibit low area specific resistance (ASR) and high oxygen exchange properties even at relatively low temperatures^{11–14}.”

Again, this is too general. While thermally activated degradation mechanisms would be alleviated by a lower operation temperature, other issues may arise. For instance, internal reforming of carbo-hydrate fuels (a major advantage of SOFCs over other fuel cells) needs a high operation temperature, and may suffer from lower operation temperature. Likewise, stainless steel interconnects suitable for 700 °C can show corrosion at intermediate temperatures.

It would be better if the authors focused on improving the performance, which directly and beneficially impacts system cost, as a driver for cell development.

Lines 56-59:

“Nanoengineered cathodes prepared by advanced thin film techniques such as pulsed laser deposition (PLD) have achieved extremely low ASR values as well as high oxygen exchange properties superior to those of conventional cathodes prepared by wet deposition techniques^{13,15,16}.”

This statement is not accurate. The best ASR value in Figure 4 is from a study by Hayd et al, who used a wet deposition technique (a sol-gel technique) for the preparation of a thin LSC electrode. Such techniques are easily scalable and much cheaper than vacuum processes such as PLD.

I suggest to sharpen the point of this sentence by using particle- or suspension-based production techniques like screen-printing as a comparison, since I suspect that the statement about nanoengineering is correct in that comparison. But there are other nanoengineering approaches (e.g. sol-gel techniques, electrostatic spray deposition) that are not suspension-based, but can be called “wet” since they are solution-based.

Thank you for pointing this out.

These parts were revised as follows:

“Toward this purpose, advanced thin film techniques such as pulsed laser deposition have been utilized to explore alternative and nanoengineered cathode materials exhibiting high performance in terms of low area specific resistance (ASR) and high oxygen exchange properties superior to those of conventional cathodes prepared by screen printing techniques^{12–17}”

Lines 63-66_

“This has the benefit of enhancing the cathode/gas and cathode/electrolyte interfacial area density and facilitating the oxygen adsorption and oxide ion transfer, respectively, leading to significant decrease of the polarization resistance values.”

It is wrong to say that perovskite/ceria composites inherently increase the cathode/gas interfacial density. This is a matter of the microstructure, not the material selection. As a matter of fact, a composite of an (mostly ORR inactive) ionic conductor and an ORR active perovskite will inherently decrease (!) the cathode/gas interfacial area, compared to a pure perovskite, when the microstructure is the same. This is trivial.

They do inherently increase the cathode/electrolyte interfacial area (again, for a given microstructure).

And so the question of whether a composite is beneficial or not becomes the question of what the rate-limiting factor is for the given cathode chemistry and microstructure. If the oxygen exchange reaction is limited by the cathode surface area, a composite will decrease performance. If it is limited by charge-transfer to the electrolyte, performance will increase.

However, doped ceria powders typically have lower sintering activities compared to LSCF or LSC. Therefore, adding ceria changes the manufacturing of composites with regard to pure

perovskites, even when the nominal production parameters (e.g. sintering temperature) remains the same. The problem in this field is related to the nature of academic publishing: when performance can be increased through fabrication of a composite, this is publishable. When performance is decreased by the composite, it is not publishable and therefore not well known.

This has been revised by removing the reference to the cathode-gas interfacial density.

Lines 70-72:

“Another critical issue is how to tailor cathode nanostructures to be gas permeable and somehow retain this type of structure even at high operating temperatures.”

It is interesting that the authors consider this a problem. Typically, gas diffusion in cathode layers is not an issue when using air as oxidant, and only becomes noticeable in electrode performance at lower pO_2 values. Furthermore, the nanostructures are very thin, so that gas diffusion polarization is likely small even for low porosity. Do the authors have some references where this has been shown to be a problem?

The concern that the structures may not remain gas-permeable at higher temperatures is also counterintuitive. As the authors show in this paper, nanostructures tend to coarsen at elevated temperatures, which actually increases porosity. The likelihood of the nanostructure sintering into a dense layer is very small, given that this is very difficult to achieve under the constraints of the rigid support.

I suspect that the authors did not mean to refer only to gas diffusion when discussing the coarsening of the nanostructures. In that case, this part should be written with improved precision.

This part has been revised as follows:

“Another critical issue is how to tailor cathode nanostructures and retain active sites for oxygen reduction even at high operating temperatures. Thin film cathodes typically suffer from an inevitable loss of nanostructures when subjected to high temperatures due to thermally induced grain coarsening²¹ and surface segregation^{22–25} occurring at such conditions, leading to significant degradation of the surface exchange properties.”

Lines 74-77:

“In addition, due to the intrinsically poor lateral conductivity of thin film cathodes, the cell performance of these materials has not yet been fully optimized in terms of appropriately selecting current collectors which would enable good electrical contact with such nanometer-sized grains^{20,21}.”

It would be good to point out here that the poor lateral conductivity is primarily a result of the columnar microstructure obtained in the gas-phase deposition process, and not the thickness. In addition, this sentence is to be understood as pre-shading the result that Pt-paste provides the best contact. It bears pointing out that whatever the best solution for contacting SOFCs is, Pt paste is not it due to the extremely high price.

This has been revised as follows:

“In addition, due to the intrinsically poor lateral conductivity of thin film cathodes arising from the characteristic columnar microstructure, the cell performance of these materials has not yet been fully optimized in terms of appropriately selecting current collectors which would enable good electrical contact with such nanometer-sized grains^{26,27}.”

Line 204:

What is a semi-open pore?

Apologies for the vague description. This has been corrected to “open pores.” (p. 9, line 210)

Lines 206-209:

“..., since the as-grown film microstructure has been initially tuned to contain porosities, this strategy seems effective in preventing the excessive grain sintering and densification which would otherwise occur for densely packed nanoscaled structures²⁶”

This paragraph is somewhat inaccurate. The primary reason why the nanostructure retains its porosity seems to be that the structure grows in distinct pillars that are isolated from each other. It is because of the lack of contact area between pillars that no densification takes place, since densification requires mass transport via diffusion, and therefore direct contact. Each individual pillar clearly shows densification at 700 °C, as shown in Figure 3. However, as the pillars densify, the distances between the pillars increase.

The increasing porosity at elevated temperature is therefore not a consequence of the presence of porosity alone, but of the particular shape of these pores..

This part has been revised as follows:

“...since the as-grown film microstructure is comprised of nanocolumns which are mostly isolated from each other, this strategy seems effective in preventing the excessive grain sintering and densification which would otherwise occur for densely packed nanoscaled structures in direct contact¹⁸.”

Lines 288-289

“Due to the high sinterability of the nanocomposite particles obtained, the fine AFL structure could be maintained even after high-temperature co-sintering.”

This is contradictory. A powder with a high sinterability would not be expected to maintain porosity after high-temperature sintering when allowed to shrink freely (as it is the case here).

As pointed out by the reviewer, the previous description was incorrect. This has been corrected to read as follows:

“Due to the low sinterability of the nanocomposite particles obtained through the spray pyrolysis method, the fine AFL structure could be maintained even after high-temperature co-

sintering.”

Line 346

“High OCV values of >1.1 V were obtained at all temperatures, attributed to the excellent integrity of the thin YSZ electrolyte and dense GDC interlayer employed in the cell architecture.”

It would be nice to simply provide the expected value of the OCV for these measurement conditions. Furthermore, the gas tightness of the setup (cell and gasket) should not be judged when using humidified fuel, but dry fuel.

This was revised as follows:

“High OCV values of ~ 1.1 V, close to the theoretical value of 1.120 V were obtained at all temperatures.”

Lines 347-350:

“The overall trend shows that employing the LSCF-GDC nanocomposite layer alone (Sample 5) is insufficient to enhance the maximum power density and moreover, it shows the lowest values among the samples at all temperatures evaluated.”

The MPD is not enhanced compared to which reference? The authors do not give any reference here.

This part has been revised to read as follows:

“The overall trend shows that employing the LSCF-GDC nanocomposite layer alone (Sample 6) is insufficient to enhance the maximum power density compared to the one combined with nanoporous LSC and moreover, it shows the lowest values among the samples at all temperatures evaluated.”

Lines 350-535

“Unlike the other two samples, the I-V dependence for this sample also shows a positive curvature, indicating that electrode activation polarization is important³⁷. This may be ascribed to the relatively dense microstructure of this layer where gas diffusion is expected to be limited.”

Here, the authors confuse activation polarization with gas diffusion polarization. The two phenomena are distinct and also result in opposite curvatures of the IV-curve.

This has been corrected to read as follows:

“The I-V dependence for this sample also shows a more prominent positive curvature at high current densities, indicating gas diffusion polarization⁴⁶. This may be ascribed to the relatively dense microstructure of this layer where gas diffusion is expected to be limited.”

- [1] D. Udomsilp et al., "Metal-Supported Solid Oxide Fuel Cells with Exceptionally High Power Density for Range Extender Systems," *Cell Reports Physical Science*, vol. 1, no. 6, p. 100072, 2020/06/24/ 2020, doi: <https://doi.org/10.1016/j.xcrp.2020.100072>.
- [2] J. Hayd, L. Dieterle, U. Guntow, D. Gerthsen, and E. Ivers-Tiffée, "Nanoscaled $\text{La}_{0.6}\text{Sr}_{0.4}\text{CoO}_{3-\delta}$ as intermediate temperature solid oxide fuel cell cathode: Microstructure and electrochemical performance," *Journal of Power Sources*, vol. 196, no. 17, pp. 7263-7270, 2011, doi: [10.1016/j.jpowsour.2010.11.147](https://doi.org/10.1016/j.jpowsour.2010.11.147).
- [3] D. Klotz, A. Weber, and E. Ivers-Tiffée, "Practical Guidelines for Reliable Electrochemical Characterization of Solid Oxide Fuel Cells," *Electrochimica Acta*, vol. 227, pp. 110-126, 2017/02/10/ 2017, doi: <https://doi.org/10.1016/j.electacta.2016.12.148>.
- [4] E. Guk, J.-S. Kim, M. Ranaweera, V. Venkatesan, and L. Jackson, "In-situ monitoring of temperature distribution in operating solid oxide fuel cell cathode using proprietary sensory techniques versus commercial thermocouples," *Applied Energy*, vol. 230, pp. 551-562, 2018/11/15/ 2018, doi: <https://doi.org/10.1016/j.apenergy.2018.08.120>.
- [5] Q. Fang, L. Blum, and D. Stolten, "Electrochemical Performance and Degradation Analysis of an SOFC Short Stack for Operation of More than 100,000 Hours," *ECS Transactions*, vol. 91, no. 1, pp. 687-696, July 10, 2019 2019, doi: [10.1149/09101.0687ecst](https://doi.org/10.1149/09101.0687ecst).

Reviewers' Comments:

Reviewer #1:

Remarks to the Author:

As a whole, I am satisfied with the response from the authors to the comments from the reviewers. IN combination with the comments from the other two reviewers's comments, the potential self-heating is still a big concern from my side. It is well known that the conductivity of YSZ is in linear response to $1000/T$ in the tested temperature range in this study. It means, a linear response of I-V characteristics. However, In some of the I-V curves of this study, a non-linear feature is observed, which could a result from the increased electrode performance at higher current density, or an increase cell temperature from self-heating. Generally for the LSC based cathode, the electrode performance is relative stable in a large range of polarization current density. It suggests the self-heating at high current density could be a contribution, at least partially. I recommend the authors to conduct additional experiment by conducting the I-V polarization both in the way of increasing current density from zero to the maximum value, and then in a reverse manner, i.e., from the highest current density to zero by step polarization, if the self heating appears, typically, higher power output would be observed when the I-V curves are measured from high polarization current to low current way. Anyway, in many of the cells as reported in literature with ultrahigh power density, self-heating could be also a problem. I recommend the authors do not to emphasize the absolute value of the power density, instead to the enhancement of the cell performance because of the special electrode structure as introduced from the PLD technique.

Once above point is reasonably response, we would suggest the publication of this manuscript in Nature Communication.

Reviewer #2:

Remarks to the Author:

Thanks for the revised manuscript. The revision is successful and I can now recommend this manuscript to be published in Nature Communications.

Reviewer #3:

Remarks to the Author:

The authors have obviously put a lot of effort into addressing the criticism of the reviewers, and have adapted a significant portion of the paper accordingly. The paper is thus improved and I would recommend it for publication

Dear Reviewers,

We would like to express our sincere thanks to all the reviewers for kindly evaluating our revised manuscript. We are very happy to receive the recommendations for publication from two of the reviewers.

Please find below the response to the remaining issue raised by Reviewer #1. We hope that the response below properly addresses the issue.

Thank you again for the opportunity to improve our contribution and we look forward to the recommendation of our article in Nature Communications.

On behalf of the co-authors,

Katherine Develos-Bagarinao

AIST, Japan

REVIEWER COMMENTS

Reviewer #1 (Remarks to the Author):

As a whole, I am satisfied with the response from the authors to the comments from the reviewers. IN combination with the comments from the other two reviewers's comments, the potential self-heating is still a big concern from my side. It is well known that the conductivity of YSZ is in linear response to $1000/T$ in the tested temperature range in this study. It means, a linear response of I-V characteristics. However, In some of the I-V curves of this study, a non-linear feature is observed, which could a result from the increased electrode performance at higher current density, or an increase cell temperature from self-heating. Generally for the LSC based cathode, the electrode performance is relative stable in a large range of polarization current density. It suggests the self-heating at high current density could be a contribution, at least partially. I recommend the authors to conduct additional experiment by conducting the I-V polarization both in the way of increasing current density from zero to the maximum value, and then in a reverse manner, i.e., from the highest current density to zero by step polarization, if the self heating appears, typically, higher power output would be observed when the I-V curves are measured from high polarization current to low current way.

Anyway, in many of the cells as reported in literature with ultrahigh power density, self-heating could be also a problem. I recommend the authors do not to emphasize the absolute value of the power density, instead to the enhancement of the cell performance

because of the special electrode structure as introduced from the PLD technique.

Once above point is reasonably response, we would suggest the publication of this manuscript in Nature Communication.

Authors' response

Dear Reviewer,

Thank you for your comments.

We understand the reviewer's concern about self-heating.

With regards to the additional experiment recommended by the reviewer, we would like to point out that a similar experiment has indeed been reported in one of the references cited in the manuscript, namely that by D. Udomsilp et al., Cell Reports Physical Science 1, 100072 (2020) <https://doi.org/10.1016/j.xcrp.2020.100072>. In the paper, they showed an apparent hysteresis behavior in the I - V characteristics (Fig. 3), which they attributed to Joule heating of the cell. Moreover, an increase in cell temperature was actually observed while *decreasing* the current, and therefore to an overestimate of the performance with this measurement. To quote directly from the article:

“For a conservative estimate of the presented performance, values for 0.7 V are extrapolated from the data points of decreasing current, whereas the values measured at 0.9 V are derived during increase of the current. This is as the cell temperature tends to further increase while decreasing the current, resulting in a steeper slope and thus in lower current densities extrapolated for 0.7 V. **In contrast, the cell temperature is closer to the setpoint during increase of the current load, whereas the progress of the curve while decreasing the current would overestimate the cell performance when relating it to the set temperature.**” (Emphasis added)

The reviewer is correct in their statement that the power output would be apparently higher when sweeping from higher to lower current load than compared to the reverse, but this will only be applicable if the extra thermal load actually becomes significant due to the increase of applied current, and it does not dissipate as fast. It can be argued that this is commonly observed for larger cells (or metal-supported cells as with the Udomsilp paper), where heat distribution is an issue and therefore invariably result to the observed hysteresis in the I - V characteristics. Furthermore, it can be seen in Fig. 3 that the hysteresis itself is dependent on the operation temperature: the higher the temperature, the less apparent the hysteresis. Self-heating is thus a complicated phenomenon that should be analyzed in view of other contributing factors such as heat dissipation, gas flow rate, conductivity of the Pt mesh, etc.

Nevertheless, as explained in our previous replies to the comments received from the other reviewers, we believe that we have already put forward sufficient arguments that whereas self-heating may occur for this type of experiments, this phenomenon is quite difficult to confirm experimentally, and within the limitations of our test bench there does not seem to be any indication of any substantial self-heating in the cells.

To reiterate the arguments presented previously:

1. Comparison of the ohmic resistance values derived from the impedance spectra at OCV and 0.75 V shows that they are not significantly different, suggesting that even if self-heating occurs under polarization, it is most likely negligible.
2. The observed non-linearity in the I - V curves is most likely attributed to a possible activation with current density. Such an anomalous behavior may be directly correlated to the complexity of the nanostructures developed in this study.

As the I - V results reported in our manuscript were all derived during increase of the current load, we believe that the values are more accurate descriptors of the actual cell performance, given that the said self-heating (if it occurs) would only overestimate the power output during decrease of the current load. It would also be worth mentioning that the configuration of cell stacks is quite different from the single button cells that we used in our experiments, thus the self-heating behavior of these two types of structures cannot be expected to be similar. Therefore, we do not see any merits for performing this additional experiment, nor will it add any significant insights relevant to the main conclusions of this study.

In the second revision of the manuscript, we have added some explanation to address the issue of self-heating on pp. 21-22.

In response to the comments received from the other reviewers, we would like to note that emphasis on the maximum power density values has already been removed in the first revised version, and only relevant values based on the current densities obtained at an operating voltage of 0.7 V were used for comparison with those reported in literature.

As a final note, we appreciate your critical assessment of our work and hope that this response satisfactorily addresses this issue for acceptance in Nature Communications.

Reviewers' Comments:

Reviewer #1:

Remarks to the Author:

I satisfy with the argument/explanations from the author as raised from my previous comments. The manuscript is now acceptable for publication in Nature Communication.